# Mining Tensor/Neuron-Level Sparsity to Maximize Mixture-of-Experts Potential in Post-Training and Inference

Weilin Cai [1]  Le Qin [1]  Shwai He [2]  Junwei Cui [1]  Ang Li [2]  Jiayi Huang [1]

## Abstract

Mixture of Experts (MoE) has emerged as a mainstream architecture for Large Language Models (LLMs), balancing computational efficiency with model scalability. While prior work has explored increasing tensor-level sparsity via finer-grained expert configurations during pre-training, we identify significant unexploited sparsity at both the tensor and neuron levels during post-training and inference. To leverage this, we propose complete expert partition for post-training and threshold-based token-expert dropping for inference. These techniques improve the Mixtral-8×7B model's average accuracy by 1% across nine downstream benchmarks (notably 4% on GSM8K). To further optimize the accuracy-efficiency trade-off for inference, we introduce dual-threshold token-expert dropping with partial expert partition and reconstruction. Our approach yields a 1.19× MoE speedup and a 0.5% accuracy gain on Mixtral-8×7B when combining post-training and inference optimizations. For inference-only optimization on OLMoE-Instruct and DeepSeek-V2-Lite-Chat, we achieve up to 1.41× MoE speedup with a negligible accuracy loss ($<0.5\%$).

## 1. Introduction

Recently, the Mixture-of-Experts (MoE) architecture (Shazeer et al., 2017; Lepikhin et al., 2021; Rajbhandari et al., 2022; Dai et al., 2024) has emerged as a mainstream design for Large Language Models (LLMs) (Cai et al., 2025; Jiang et al., 2024; Muennighoff et al., 2025; Liu et al., 2024a), primarily due to its superior trade-off between computational efficiency and model quality. This efficiency stems from the tensor-level sparsity inherent in the MoE architecture, which can be conceptually viewed as partitioning a large Feed-Forward Network (FFN) into fine-grained sub-FFNs (termed "experts") and selectively activates only a subset of these experts to process each input token.

During pre-training, existing research pursues higher tensor-level sparsity through finer-grained expert designs (Dai et al., 2024; He, 2024; Abnar et al.; Ludziejewski et al., 2024), while maintaining sufficient compute intensity to fully utilize hardware bandwidth. While this architectural design endows the model with a superior accuracy-efficiency trade-off, our analysis of pre-trained MoE models reveals that this sparsity is significantly under-exploited during post-training and inference. As deployment conditions and workloads differ significantly from the pre-training scenarios, static expert configurations established during pre-training fail to capture additional sparsity opportunities available during these subsequent stages.

Motivated by the observations, we introduce complete expert partition for the post-training stage and threshold-based token-expert dropping for the inference stage. Validated by empirical results and behavioral analysis, these methods effectively leverage latent tensor-level sparsity to improve downstream accuracy. Specifically, complete expert partition converts a pre-trained MoE model into a configuration with finer-grained experts. This transformation preserves mathematical consistency, amplifying accuracy gains during supervised fine-tuning. Moreover, during inference, threshold-based token-expert dropping (denoted as "1T-Drop") filters MoE computation noise via normalized gating scores to improve accuracy.

Furthermore, we identify a pronounced activation imbalance at both the tensor and neuron levels (Figure 1), where the output of each FFN neuron is modulated by the product of its gating score and activation magnitude. To exploit this dual-sparsity (comprising tensor-level and neuron-level sparsity), we propose dual-threshold token-expert dropping with partial expert partition and reconstruction (collectively denoted as "2T-Drop"). This approach incorporates three key designs: (1) *Static expert partition and reconstruction*, which divides neurons in each expert into major and minor sub-experts based on importance profiling from cal-

[1]The Hong Kong University of Science and Technology (Guangzhou), Guangzhou, China [2]University of Maryland, College Park, Maryland, USA. Correspondence to: Jiayi Huang <hjy@hkust-gz.edu.cn>.

*Proceedings of the 43rd International Conference on Machine Learning*, Seoul, South Korea. PMLR 306, 2026. Copyright 2026 by the author(s).

ibration samples; (2) *Dynamic token-expert computation dropping*, which selectively skips computations via dual-thresholding of normalized gating scores, applying lower and upper thresholds to major and minor sub-experts, respectively; (3) *Load-aware thresholding*, which dynamically adjusts thresholds according to workload imbalances across devices, thereby reducing the drop rate and preserving accuracy while maintaining speedup benefits in expert-parallel (EP) deployment.

In our experiments, we apply complete expert partition to transform the Mixtral-8×7B model from 8 experts into 32 finer-grained experts, reducing fine-tuning loss and improving downstream average accuracy by 0.6%. Further integration of 1T-Drop with $T_{drop}^1 = 0.05$ during inference increases this gain to 0.9%. Alternatively, when employing 2T-Drop to achieve a better accuracy-efficiency trade-off during inference, our approach yields a 1.19× MoE speedup and a 0.5% accuracy gain on Mixtral-8×7B.

Moreover, when directly applying 2T-Drop as an inference-only optimization without fine tuning, an approximate 25% drop rate reduces average benchmark accuracy by only 0.08%–0.28% across three MoE models. Notably, nearly all drop rates translate directly into proportional computation reduction and speedup, a result difficult to achieve with other sparsity-based acceleration techniques. With load-aware thresholding in EP, our method achieves a 1.41× MoE speedup with only 0.5% average accuracy loss.

In summary, our contributions are as follows:

- We identify and characterize dual sparsity within MoE architectures at both the tensor and neuron levels, demonstrating their potential to enhance performance during post-training and inference stages.

- We propose complete expert partition for the post-training and threshold-based token-expert dropping for inference. These methods can synergize to enhance tensor-level sparsity, leading to improved accuracy.

- We design dual-threshold token-expert dropping with partial expert partition and reconstruction, which leverages both tensor- and neuron-level sparsity to improve inference efficiency while maintaining model accuracy.

- We conduct extensive experiments and analyses to show the effectiveness of our proposed approaches.

## 2. Preliminary

By visualizing MoE activation patterns during inference with the pre-trained OLMoE model (Muennighoff et al., 2025), as shown in Figure 1, we observe that the output of MoE module is governed by dual sparsity at both the tensor and neuron levels. Specifically, color variations across rows (y-axis) reflect tensor-level sparsity, while color differences among points within each row (x-axis) capture neuron-level.

### 2.1. Tensor-Level Sparsity in Mixture of Experts

MoE (Shazeer et al., 2017; Lepikhin et al., 2021; Fedus et al., 2022) is a neural network architecture that dynamically selects experts to process each token. An MoE layer comprises $E$ expert networks alongside a gating network $G$. The gating network, typically a linear network with a softmax activation function, calculates a selection probability (gating score **s**) for each expert, defined as:

$$\mathbf{s} = G(\mathbf{x}) = \text{Softmax}(\mathbf{x} \cdot \mathbf{W}_g), \qquad (1)$$

where $\mathbf{W}_g \in \mathbb{R}^{d_{model} \times E}$ is the weight matrix of the linear gating network. Based on the gating scores, the Top-$K$ gating method is commonly used to route each input token to a subset of experts for computation, defined as:

$$g_e(\mathbf{x}) = \begin{cases} \mathbf{s}_i & \text{if } i \in \text{TopK}(\mathbf{s}, K), \\ 0 & \text{otherwise,} \end{cases} \qquad (2)$$

where $g_e(\mathbf{x})$ denotes the gating score for expert $e$. Each input token is processed by the $K$ experts with the highest gating scores, and the MoE output is a weighted sum of the outputs of the selected experts:

$$\mathbf{y} = \sum_{e=1}^{E} g_e(\mathbf{x}) \cdot f_e(\mathbf{x}), \qquad (3)$$

where $f_e(\mathbf{x})$ denotes the output of expert $e$. Using SwiGLU (Shazeer, 2020) feed-forward network (FFN) expert as an example, the expert output $f(\mathbf{x})$ is formulated as:

$$f(\mathbf{x}) = (\text{Swish}(\mathbf{x} \cdot \mathbf{W}_1) \odot (\mathbf{x} \cdot \mathbf{W}_3)) \cdot \mathbf{W}_2, \qquad (4)$$

where $\mathbf{W}_1, \mathbf{W}_3 \in \mathbb{R}^{d_{model} \times d_{ffn}}$ and $\mathbf{W}_2 \in \mathbb{R}^{d_{ffn} \times d_{model}}$ denote FFN weights, and the Swish activation function (Ramachandran et al., 2017) is used. The three linear transformations associated with $\mathbf{W}_1, \mathbf{W}_3, \mathbf{W}_2$ are referred to as the gate, up, and down projections, respectively.

Recent studies (Dai et al., 2024; He, 2024; Abnar et al.; Ludziejewski et al., 2024) have systematically demonstrated that configuring experts at finer granularity—while maintaining a fixed per-token computational budget—can substantially reduce pre-training loss. For instance, an MoE model with 32 experts of intermediate size 1024 and Top-8 selection outperforms an MoE model with 8 experts of intermediate size 4096 and Top-2 selection. However, further increasing the number of experts may reduce computational efficiency of training due to lower compute intensity and increased gating overhead.

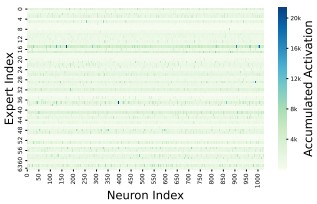

*Figure 1.* Visualization of accumulated absolute activation values for each neuron across 64 SwiGLU FFN experts in a single MoE layer OLMoE (Muennighoff et al., 2025) model during inference, highlighting tensor-level sparsity (y-axis) and neuron-level sparsity (x-axis) inherent to MoE architectures.

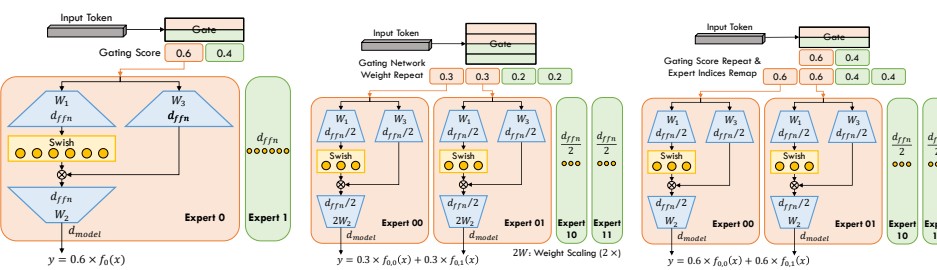

(a) Original MoE Layer      (b) Complete Expert Partition      (c) Partial Expert Partition

*Figure 2.* Illustration of expert partition methods, demonstrated by transforming a pre-trained 2-expert MoE model into a finer-grained 4-expert model. (a) Original MoE layer in the pre-trained model. (b) **Complete transformation**, which involves repeating the gating network weights, partitioning expert neurons, and scaling the down-projection weights $\mathbf{W}_2$. (c) **Partial transformation**, which involves partitioning expert neurons, repeating gating scores, and remapping expert indices.

## 2.2. Neuron-Level Sparsity in Mixture of Experts

Beyond the tensor-level sparsity inherent to the MoE architectures, prior research has investigated computation dropping and parameter pruning upon weight sparsity (Frantar & Alistarh, 2023; Sun et al., 2024; Fan et al., 2025) and activation sparsity (Zheng et al., 2023; Song et al., 2024) in dense FFN. However, these approaches face three primary challenges: (1) **Accuracy Sensitivity:** LLMs exhibit low tolerance for high dropping or pruning rates, leading to significant accuracy degradation as these rates increase; (2) **Hardware Inefficiency:** Highly fine-grained sparsity resulting from uncoordinated dropping is difficult to translate into wall-clock speedups due to the limited support for irregular, unstructured computation patterns in existing hardware and kernel designs; and (3) **Activation Dependency:** Most methods focus primarily on the ReLU activation function (Agarap, 2018), which naturally produces zeros, and therefore cannot be directly applied to modern LLMs employing SwiGLU activations (Shazeer, 2020). In this work, we identify the activation sparsity present in MoE FFN experts as neuron-level sparsity and propose a framework that coordinates it with tensor-level sparsity. By jointly leveraging these two complementary forms of sparsity, we address the aforementioned challenges and improve both algorithmic accuracy and system efficiency.

## 3. Methodology

This section details our proposed methods for achieving accuracy-oriented improvements (Section 3.1, Section 3.2) and optimizing the accuracy-efficiency trade-off (Section 3.3). We present the methods progressively across two application stages:

- **Post-training** (Section 3.1): Complete Expert Partition.
- **Inference** (Section 3.2, Section 3.3): 1T-Drop → 2T-Drop (Partial Expert Partition and Expert Reconstruction with Neuron Profiling) → Load-Aware Thresholding in EP.

These components can be integrated into a unified pipeline to achieve superior overall performance.

## 3.1. Complete Expert Partition for Post-Training

As discussed in Section 2.1, prior work has demonstrated that increasing tensor-level sparsity during pre-training—by configuring finer-grained experts—can enhance model quality. In this work, we investigate the potential for performance gains by partitioning the experts of a pre-trained model into $P$ finer-grained tensors while keeping the total activated parameter volume constant. To this end, we propose complete expert partition, as illustrated in Figure 2(b), which ensures that the transformed MoE model maintains mathematical equivalence to the original model.

Regarding the gating network weights ($\mathbf{W}_g$), complete expert partition replicates each gating embedding ($h_e$) $P$ times to construct the expanded gating weights ($\mathbf{W}_g^P$). While this replication preserves the original logits ($l_{e,p} = l_e$), the softmax normalization over the $E \times P$ experts scales each finer-grained gating score to $s_{e,p} = \frac{s_e}{P}$. Given that the sum of the partitioned expert outputs $\sum_{p=1}^{P} f_{e,p}(\mathbf{x}_i)$ equals the original output $f_e(\mathbf{x}_i)$, the output $\mathbf{y}_i^P$ can be as:

$$
\begin{aligned}
\mathbf{y}_i^P &= \sum_{e=1}^{E} \sum_{p=1}^{P} \frac{1}{P} \cdot \frac{\exp(l_{e,p})}{\sum_{j=1}^{E} \sum_{k=1}^{P} \exp(l_{j,k})} \cdot f_{e,p}(\mathbf{x}_i) \\
&= \frac{1}{P} \sum_{e=1}^{E} \cdot \frac{\exp(l_e)}{\sum_{j=1}^{E} \exp(l_j)} \cdot \sum_{p=1}^{P} f_{e,p}(\mathbf{x}_i) = \frac{\mathbf{y}_i}{P}.
\end{aligned}
\tag{5}
$$

To ensure that $\mathbf{y}_i^P$ is equivalent to the original output $\mathbf{y}_i$, the result must be scaled by a factor of $P$. There are two ways to achieve this: (1) multiplying the gating scores by $P$, or (2) scaling the expert weight $\mathbf{W}_2$ of the down-projection by $P$. To preserve the original model structure without modifying the existing framework, we choose to scale the expert weights for complete transformation. A more detailed derivation is provided in Appendix A.1.

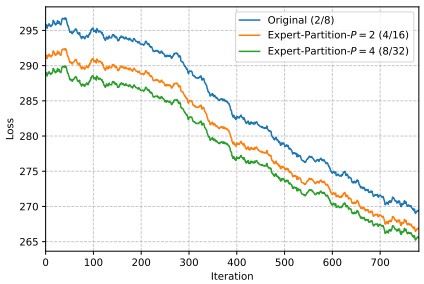

*Figure 3.* Fine-tuning loss curves for Mixtral-8×7B (Jiang et al., 2024) models under different configurations, including the original model (activating top-2 out of 8 experts) and models completely transformed with partitioned experts (activating top-4 out of 16 experts and top-8 out of 32 experts).

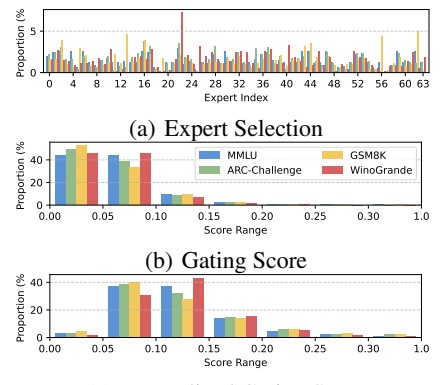

*Figure 4.* Distributions of three metrics observed during OLMoE model inference across four distinct benchmark tasks.

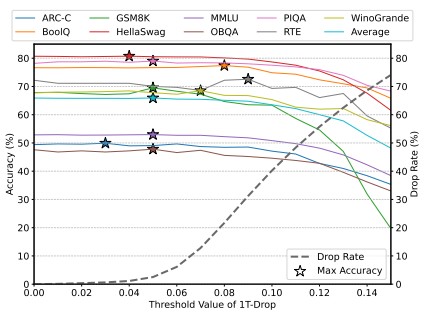

*Figure 5.* Benchmark accuracy and token-expert computation drop rate for OLMoE model inference using different threshold values of 1T-Drop. "Stars" indicate the threshold that achieves the maximum accuracy for each benchmark.

As shown in Figure 3, models with partitioned experts exhibit significantly lower fine-tuning loss curves compared to the original Mixtral-8×7B (Jiang et al., 2024) model; moreover, finer-grained experts yield further loss reduction. However, increasing the number of partitions beyond a certain point offers only marginal improvements. Experimental results on the downstream tasks, presented in Table 1 and discussed in Section 4.2, further confirm the accuracy gains.

### 3.2. Threshold Token-Expert Dropping for Inference

Motivated by the correlation between tensor-level sparsity and the superior accuracy-efficiency trade-off offered by MoE architectures, we analyze the behavior of the gating mechanism to identify characteristics that can be leveraged for inference optimization. We first observe the dynamic imbalances in expert activation across various tasks (Figure 4(a)), suggesting that static expert pruning approaches (Lu et al., 2024; Chen et al., 2025; Kim et al., 2021; Koishekenov et al., 2023) may suffer from poor generalization and accuracy degradation.

Furthermore, our investigation reveals a consistent phenomenon within gating mechanism: the distribution of gating scores for activated token-expert pairs remains remarkably stable across diverse downstream tasks. As shown in Figure 4(b), the raw gating scores exhibit a heavy-tailed distribution, where the majority of scores are concentrated in the lower intervals ([0, 0.05] and [0.05, 0.1]), with a rapid decay in frequency as the score magnitude increases. Normalizing the Top-$K$ gating scores, as shown in Figure 4(c), yields a flatter distribution while preserving the cross-task consistency observed in the raw scores.

From a signal-to-noise perspective, Top-$K$ expert selection acts as a filter, reducing the noise inherent in raw gating scores. When examining normalized gating scores, this noise becomes distinct, manifesting as a small portion of

low-magnitude values seemingly restricted to the [0, 0.05].

To leverage this insight, we propose an operation termed "1T-Drop," which selectively drops token-expert computations whose normalized gating scores fall below a specified threshold ($T_{drop}^1$). This can be formulated as:

$$\mathbf{y} = \underbrace{\sum_{i \in \mathcal{S}} g_i f_i(\mathbf{x})}_{\text{Signal } (\hat{g}_i \geq T_{drop}^1)} + \underbrace{\sum_{j \in \mathcal{N}} g_j f_j(\mathbf{x})}_{\text{Noise } (\hat{g}_i < T_{drop}^1)}, \quad (6)$$

where $\hat{g}$ denotes the normalized gating score. It is worth noting that for some MoE models (Liu et al., 2024b; Yang et al., 2025) already normalize the gating scores of activated experts, additional normalization step is unnecessary.

Our empirical results show that applying a low threshold (approximately 0.05) for dropping computations can actually improve accuracy, as shown in Figure 5. Across all benchmarks, peak accuracy is achieved when certain token-expert computations are dropped, suggesting that computations with very low gating scores ([0, 0.05]) may negatively impact overall performance, a finding that aligns with our hypothesis regarding noise reduction.

### 3.3. Dual-Threshold Token-Expert Dropping with Partial Expert Partition and Reconstruction

While directly dropping token-expert computations based on a single gating threshold (1T-Drop) can provide significant speedups at higher thresholds, it often leads to accuracy degradation due to the loss of critical information. To mitigate this, we propose 2T-Drop, a hierarchical dropping strategy that exploits the observed dual sparsity (Figure 1) within MoE module. Specifically, 2T-Drop achieves tensor-level differentiation through partial expert partition (Figure 2(c)) while simultaneously coordinating neuron-level architectural refinement within the partitioned tensors. It creates

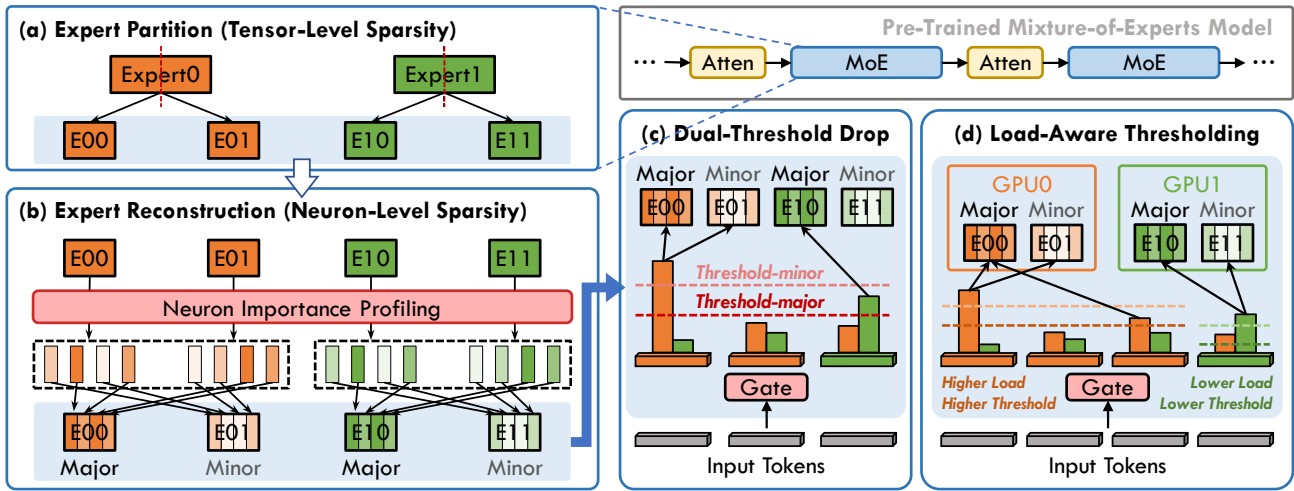

*Figure 6.* Overview of the proposed dual-threshold token-expert computation dropping approach (2T-Drop) and its enhancement through load-aware thresholding under expert parallelism, in the context of deploying pre-trained MoE models for inference.

a tiered computation mode that preserves high-impact features while aggressively dropping redundant computation. As illustrated in Figure 6, this strategy is realized through an integrated pipeline of:

**(a) Partial Expert Partition.** We employ the partial transformation of expert partition (Figure 2(c)) to enhance tensor-level sparsity. This approach enables finer-grained and more flexible combinations of token-expert computation dropping at the tensor level. Unlike the complete transformation, the partial variant preserves the original gating network and modifies only the Top-$K$ selection results through two operations: (1) repeating the gating scores and (2) remapping the expert indices (see Appendix A.2 for further details). This design is specifically intended to minimize gating overhead for extreme inference efficiency, though it necessitates specialized optimizations within the inference framework to handle the modified index mapping.

**(b) Expert Reconstruction.** To exploit neuron-level sparsity within each expert, we perform neuron importance profiling on calibration samples. Neurons are then reorganized to reconstruct a major sub-expert comprising higher-importance neurons and a minor sub-expert comprising lower-importance neurons. In our implementation, expert partitioning and reconstruction are executed as a unified process: all neurons in an original expert are profiled and reorganized into these two distinct sub-experts. This static approach leverages neuron-level sparsity while avoiding the substantial overhead of dynamically identifying neuron activations for dropping at runtime. Furthermore, we experiment with various neuron importance profiling methods within SwiGLU experts:

(1) Accumulated Gate Value:

$$\text{Importance} = \sum \text{Swish}(\mathbf{x} \cdot \mathbf{W}_1^{\text{neuron}}), \qquad (7)$$

(2) Accumulated Absolute Gate Value:

$$\text{Importance} = \sum \left| \text{Swish}(\mathbf{x} \cdot \mathbf{W}_1^{\text{neuron}}) \right|, \qquad (8)$$

(3) Accumulated Gate-Up Value:

$$\text{Importance} = \sum (\text{Swish}(\mathbf{x} \cdot \mathbf{W}_1^{\text{neuron}}) \odot (\mathbf{x} \cdot \mathbf{W}_3^{\text{neuron}})), \quad (9)$$

(4) Accumulated Absolute Gate-Up Value:

$$\text{Importance} = \sum \left| \text{Swish}(\mathbf{x} \cdot \mathbf{W}_1^{\text{neuron}}) \odot (\mathbf{x} \cdot \mathbf{W}_3^{\text{neuron}}) \right|. \qquad (10)$$

Here, $\mathbf{W}_1^{\text{neuron}}$ and $\mathbf{W}_3^{\text{neuron}}$ denote each neuron's $\mathbf{W}_1$ and $\mathbf{W}_3$ weights, following the formulation of the SwiGLU expert in Equation (4). Empirically, we observe that different models exhibit varying affinities for these profiling methods, highlighting the need to empirically determine the optimal configuration for each specific model.

**(c) Dual-Threshold Drop.** Building on the reconstructed minor and major sub-experts, we propose the dual-threshold drop (2T-Drop) method. Unlike binary dropping, this approach applies a tiered thresholding logic: a higher threshold-minor ($T_{minor}^2$) for minor sub-experts and a lower threshold-major ($T_{major}^2$) for major sub-experts. Specifically, experts with gating scores above $T_{minor}^2$ are fully engaged in computation, while those with gating scores below $T_{major}^2$ are entirely dropped, similar to the 1T-Drop method. Uniquely, experts with gating scores between $T_{minor}^2$ and $T_{major}^2$ compute only the major sub-experts. This "partial execution" state preserves critical neuron-level features that would otherwise be lost in a single-threshold system. This can be formulated as

$$\mathbf{y} = \sum_{e=1}^{E} g_e \left[ \mathbb{I}_{\hat{g}_e \geq T_{major}^2} f_e^{\text{maj}}(\mathbf{x}) + \mathbb{I}_{\hat{g}_e \geq T_{minor}^2} f_e^{\text{min}}(\mathbf{x}) \right]. \quad (11)$$

Based on our empirical experiments, we select dual thresholds of $T^2_{major} = T^1_{drop} - 0.01$ and $T^2_{minor} = T^1_{drop} + 0.01$, which preserve a drop rate similar to 1T-Drop while achieving higher inference accuracy by selectively retaining high-importance neurons in marginal experts.

**(d) Load-Aware Thresholding.** Load imbalance among distributed devices is a major factor limiting efficiency in inference with EP. Since the overall execution is bottlenecked by the device with the heaviest computational load, a uniform dropping strategy across all devices unnecessarily degrade accuracy on those with lighter workloads without providing any additional speedup. To address this, we propose a load-aware thresholding mechanism that dynamically adjusts the dropping rate based on individual device load. This approach enables the system to adaptively balance computation across devices while maintaining high accuracy.

As shown in Figure 6(d), we employ a step-down thresholding strategy: devices with higher workloads apply higher thresholds to dropping more computations, while devices with lighter workloads use lower thresholds to preserve more information. To minimize control overhead in distributed environments, we calculate the ratio of the actual load to the ideal balanced load for each device. If this ratio exceeds 1, the threshold is set to a predefined maximum; if it is below 1, the threshold is proportionally reduced according to the deviation from the ideal load. This method ensures that all devices minimize computation dropping while maintaining their individual load at or below that of the originally most-loaded device.

In addition, since finer-grained partitioning can reduce arithmetic intensity and lead to suboptimal GPU utilization, we choose to partition and reconstruct each original expert into only two sub-experts (major and minor). To enhance efficiency, we optimize the corresponding Triton kernel to accommodate the modified computation granularity of the token-expert grouped-GEMM and the additional control operations in the gating function.

# 4. Evaluation

## 4.1. Experimental Setup

To evaluate the efficacy of our proposed methods, we conduct experiments on a server equipped with 8 Nvidia H20 GPUs. Specifically, we utilize EleutherAI's LM-Evaluation-Harness (Gao et al., 2024) and OpenCompass (Contributors, 2023) to assess model quality, reporting either accuracy or normalized accuracy for each benchmark, as applicable. Our evaluation tasks include zero-shot evaluations on the ARC-C (Clark et al., 2018), BoolQ (Clark et al., 2019), HellaSwag (Zellers et al., 2019), MMLU (Hendrycks et al., 2021), OBQA (Mihaylov et al., 2018), PIQA (Bisk et al., 2020), RTE (Wang et al., 2018), WinoGrande (Sakaguchi

et al., 2021), HumanEval (Chen et al., 2021), and MATH500 (Lightman et al., 2024) benchmarks, as well as 5-shot evaluation on GSM8K (Cobbe et al., 2021). We use RULER (Hsieh et al., 2024) to evaluate the impact across different context lengths. We utilize the Tulu-3-sft-mixture dataset (Lambert et al., 2024) for our fine-tuning experiments. We implement our proposed inference optimizations and evaluate its acceleration effectiveness upon the SGLang (Zheng et al., 2024) , which supports efficient distributed inference for prevailing MoE models such as Mixtral (Jiang et al., 2024), OLMoE (Muennighoff et al., 2025), and DeepSeek (Liu et al., 2024a), GPT-OSS-20B (Agarwal et al., 2025) and Qwen3-30B-A3B (Yang et al., 2025).

## 4.2. Accuracy-Oriented Improvements

We conduct experiments to validate the accuracy benefits of promoting tensor-level sparsity via complete expert partition during post-training and threshold-based token-expert dropping (1T-Drop) during inference.

**Expert Partition.** We apply complete expert partition to the Mixtral-8×7B model, partitioning its original 8 experts into 16 ($P = 2$) and 32 ($P = 4$) finer-grained experts. As shown in Table 1, the partitioned models initially retain the same downstream accuracy with only negligible fluctuations. This consistency is attributed to the mathematical equivalence maintained by the partitioning process, though minor variations may arise due to floating-point precision errors. However, the benefits become evident after fine-tuning. Models with partitioned experts not only exhibit significantly lower fine-tuning loss curves (Figure 3) but also achieve higher downstream accuracy (Table 1). Specifically, while fine-tuning improves the original model's (2/8) average accuracy from 70.18% to 70.53%, the $P = 2$ configuration amplifies this to 70.86%, and the $P = 4$ configuration further increases it to 71.12%.

**Expert Partition + 1T-Drop.** Building on the gains from fine-tuning, the 8/32 ($P = 4$) model achieves a further improvement in average accuracy to 71.43% (including a notable 4% increase on GSM8K) when applying 1T-Drop with a threshold of 0.05 to reduce MoE noise.

## 4.3. Improving the Accuracy-Efficiency Trade-off

Given that computation dropping is an effective method for balancing accuracy and efficiency, we evaluate this trade-off from both perspectives in the following subsections.

### 4.3.1. INTEGRATION OF POST-TRAINING AND INFERENCE IMPROVEMENTS

As shown in Table 1, 2T-Drop with neuron-level reconstruction achieves a higher average accuracy than 1T-Drop and 2T-Drop without neuron-level reconstruction, while achiev-

*Table 1.* Comparison of downstream accuracy between the original Mixtral-8×7B model and its expert-partitioned variant. 2T (Partition) denotes the 2T-Drop without neuron-level reconstruct. Note that setting $T^2_{major} = T^2_{minor}$ is equivalent to using 1T-Drop with $T^1_{drop}$.

| Model | E-Activ./Total | Drop Method | $T^2_{major}$ | $T^2_{minor}$ | Drop Rate | ARC-C | BoolQ | GSM8K | HellaSwag | MMLU | OBQA | PIQA | RTE | WinoGrande | AVG.(↑) |
|---|---|---|---|---|---|---|---|---|---|---|---|---|---|---|---|
| | 2/8 | - | - | - | 0 | 59.47 | 85.14 | 58.07 | 84.05 | 67.13 | 47.00 | 83.79 | 70.40 | 76.56 | 70.18 |
| Mixtral-8×7B | 4/16 ($P=2$) | - | - | - | 0 | 59.56 | 85.32 | 58.30 | 84.02 | 67.05 | 47.20 | 83.41 | 70.76 | 76.01 | 70.18 |
| | 8/32 ($P=4$) | - | - | - | 0 | 59.47 | 85.26 | 58.07 | 83.99 | 67.22 | 46.80 | 83.46 | 70.76 | 76.72 | 70.19 |
| | 2/8 | - | - | - | 0 | 60.58 | 87.06 | 60.73 | 82.99 | 64.92 | 46.20 | **83.62** | 71.84 | 76.87 | 70.53 |
| Fine-Tuned Mixtral-8×7B Accuracy-Oriented | 4/16 ($P=2$) | - | - | - | 0 | 59.56 | 87.06 | 62.85 | 82.96 | **65.65** | 47.00 | 83.3 | 72.92 | 76.48 | 70.86 |
| | 8/32 ($P=4$) | - | - | - | 0 | 60.67 | 87.55 | 62.85 | 83.06 | 65.10 | **47.60** | 83.46 | 72.92 | 76.87 | 71.12 |
| | 8/32 ($P=4$) | 1T-Drop | 0.05 | 0.05 | 9.0% | 60.49 | 87.55 | **64.52** | 83.14 | 65.23 | 47.60 | 83.19 | 74.01 | 77.11 | **71.43** |
| | 2/8 | 1T-Drop | 0.30 | 0.30 | 20.3% | 59.39 | 87.06 | 61.84 | 82.72 | 64.26 | 46.40 | 82.86 | 71.48 | 76.64 | 70.29 |
| Fine-Tuned Mixtral-8×7B Accuracy-Efficiency Trade-off | 4/16 ($P=2$) | 1T-Drop | 0.15 | 0.15 | 21.0% | 59.64 | 87.00 | 63.46 | 82.58 | 64.75 | 46.40 | 83.13 | 73.65 | 76.24 | 70.76 |
| | 8/32 ($P=4$) | 1T-Drop | 0.08 | 0.08 | 23.9% | **59.73** | 87.31 | 62.85 | 82.75 | 64.76 | 47.00 | 83.03 | 74.01 | 76.48 | 70.88 |
| | 8/32 ($P=4$) | 2T (Partition) | 0.07 | 0.09 | **24.0%** | 59.47 | 87.25 | 63.15 | **82.90** | 64.62 | 47.00 | **83.51** | 74.37 | 75.85 | 70.90 |
| | 8/32 ($P=4$) | 2T (Reconstruct) | 0.07 | 0.09 | **24.0%** | 58.79 | **87.40** | **63.61** | 82.26 | **64.78** | 47.60 | 82.86 | **74.73** | 77.03 | **71.04** |

*Table 2.* Comparison of downstream accuracy across different drop methods evaluated on two open-source instruction-finetuned models in inference-only scenario. Note that setting $T^2_{major} = T^2_{minor}$ is equivalent to using 1T-Drop with $T^1_{drop}$.

| Model | Drop Method | $T^2_{major}$ | $T^2_{minor}$ | Drop Rate | ARC-C | BoolQ | GSM8K | HellaSwag | MMLU | OBQA | PIQA | RTE | WinoGrande | AVG.(↑) |
|---|---|---|---|---|---|---|---|---|---|---|---|---|---|---|
| | No Drop | - | - | 0 | 49.40 | 76.64 | 67.85 | 80.70 | 52.86 | 47.60 | 78.18 | 72.20 | 67.72 | 65.91 |
| OLMoE-Instruct | 1T-Drop | 0.08 | 0.08 | 21.7% | 48.46 | **77.40** | 64.67 | 80.11 | 52.23 | 45.60 | 78.35 | **72.20** | 66.85 | 65.10 |
| | 2T (Reconstruct) | 0.07 | 0.09 | 22.0% | **50.00** | 77.00 | 67.22 | **80.30** | 52.45 | 47.80 | **79.38** | 71.12 | 65.59 | **65.63** |
| | No Drop | - | - | 0 | 53.92 | 82.91 | 65.05 | 80.81 | 56.91 | 45.20 | 81.12 | 72.56 | 71.98 | 67.83 |
| DeepSeek-V2-Lite-Chat | 1T-Drop | 0.12 | 0.12 | 27.0% | 51.79 | 82.91 | 63.61 | 80.30 | 55.18 | 44.40 | **81.61** | 74.37 | 71.27 | 67.27 |
| | 2T (Reconstruct) | 0.11 | 0.13 | 26.9% | **52.47** | **82.94** | 64.37 | **80.37** | 55.58 | 44.80 | 81.39 | **74.73** | 72.22 | **67.65** |

ing a similar computation drop rate. Notably, compared to the original fine-tuned model (2/8) without computation dropping, 2T-Drop(Reconstruct) achieves a 1.19× MoE speedup (Figure 7), driven by a 24% reduction in computation, while achieving a 0.5% higher average accuracy.

### 4.3.2. INFERENCE-ONLY IMPROVEMENTS

Since 1T-Drop and 2T-Drop can be employed as inference-only optimizations, we evaluate them on the open-source OLMoE-Instruct and DeepSeek-V2-Lite-Chat models (Table 2). The partial results in Table 1 can also be interpreted as conducting inference-only optimizations on our fine-tuned Mixtral-8×7B (8/32) model.

**Impact on Accuracy.** As shown in Table 1 and Table 2, applying 1T-Drop for MoE computation dropping on the evaluated models leads to significant accuracy degradation, while applying 2T-Drop with only expert partitioning results in a similar level of accuracy loss. In contrast, when expert partitioning is combined with the reconstruction of major and minor sub-experts, 2T-Drop substantially minimizes accuracy loss at the same drop rate.

Specifically, imposing an approximate 25% drop rate yields only a 0.08% reduction in average accuracy for Mixtral (compared to the 71.12% baseline of the fine-tuned Mixtral-8×7B (8/32)), 0.28% for OLMoE, and 0.18% for DeepSeek. Note that because the DeepSeek-V2-Lite-Chat model utilizes a shared expert architecture, its drop rate is calculated as the ratio of dropped routed expert computations to the total routed and shared expert computations. Additional evaluations on Qwen3-30B-A3B and GPT-OSS-20B are presented in Appendix D and Table 5.

Moreover, in some tasks, the drop methods outperform the

*Table 3.* Comparison of downstream accuracy between "no drop" and "2T-Drop" evaluated on challenging generation and long-context benchmarks with context lengths up to 16K.

| Model | Drop Method | HumanEval | MATH500 | RULER 4K | RULER 8K | RULER 16K | AVG.(↑) |
|---|---|---|---|---|---|---|---|
| OLMoE | No Drop | 34.8 | 19.4 | 51.6 | – | – | 35.3 |
| | 2T-Drop | 35.4 | 19.8 | 50.2 | – | – | 35.1 |
| DeepSeek | No Drop | 48.2 | 22.4 | 85.9 | 74.9 | 70.8 | 60.4 |
| | 2T-Drop | 47.7 | 22.1 | 85.6 | 73.1 | 73.0 | 60.3 |

baseline without dropping. This phenomenon may be attributed to the factors discussed in Section 3.2, where applying an appropriate dropping threshold reduces noise, thereby enhancing accuracy. Furthermore, the impact of different drop methods on accuracy varies across tasks.

Additionally, We evaluate 2T-Drop on more challenging generation and long-context benchmarks with context lengths up to 16K, following the same 2T-Drop configuration in Table 2. As shown in Table 3, 2T-Drop maintains performance comparable to the no-drop baseline across both models. For OLMoE, whose `max_position_embeddings` is limited to 4K, evaluations beyond 4K suffer from severe accuracy collapse and are therefore not meaningful; we thus omit its 8K and 16K results.

**Efficiency Improvement.** Following the analysis of MoE computation drop rates in Table 1 and Table 2, we assess their impact on real-world throughput across different models and deployment scenarios. We consider a wide range of strategies, including single-GPU and multi-GPU environments utilizing TP or EP, to validate the versatility of our methods. Our results demonstrate consistent efficiency gains across all tested settings, directly attributable to the reduced computational requirements of our approach.

Specifically, the observed drop rates (22%—27%) yield sig-

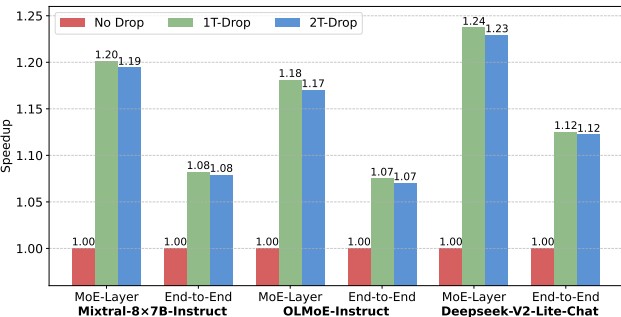

*Figure 7.* Comparison of the actual speedups achieved by 1T-Drop and 2T-Drop with the drop rates reported in Table 2. Specifically, Mixtral is deployed with TP=8 on an 8×H20 node, OLMoE is deployed on a single H20 GPU, and DeepSeek is deployed with EP=8 on an 8×H20 node.

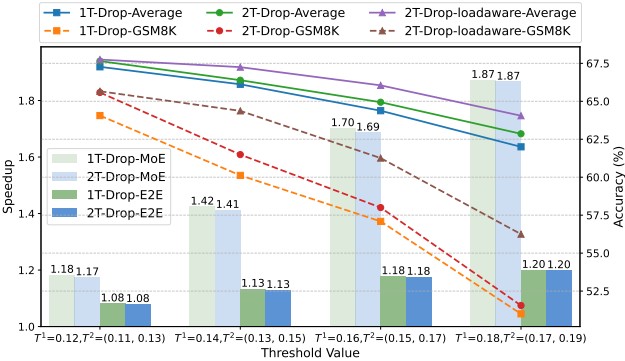

*Figure 8.* Comparison of speedup and accuracy among 1T-Drop, 2T-Drop, and 2T-Drop with load-aware thresholding for DeepSeek-V2-Lite-Chat model inference on an 8×H20 node with EP=8. $T^1$ represents the threshold applied in 1T-Drop, while $T^2$ denotes the thresholds utilized in 2T-Drop.

nificant speedups for the MoE module (1.17×—1.23×) and end-to-end improvements (1.07×—1.12×). Unlike traditional sparsity-based methods that require specialized hardware to be effective at low sparsity levels, our tensor-level dropping mechanism is inherently well-suited for existing hardware architectures. Additionally, through the use of optimized kernels, 2T-Drop achieves speedup parity with 1T-Drop, effectively mitigating the overhead of its finer-grained dropping granularity.

### 4.3.3. LOAD-AWARE THRESHOLDING IMPROVEMENT

As shown in Figure 8, increasing the drop threshold improves acceleration at the cost of accuracy. The results show that 2T-Drop outperforms 1T-Drop, and its accuracy is further enhanced by load-aware thresholding. Specifically, with the integration of load-aware thresholding, 2T-Drop achieves a 1.41× MoE speedup and a 1.13× end-to-end speedup with a 0.5% average accuracy loss. For real-world inference, the drop threshold can be dynamically tuned to meet specific throughput or accuracy targets.

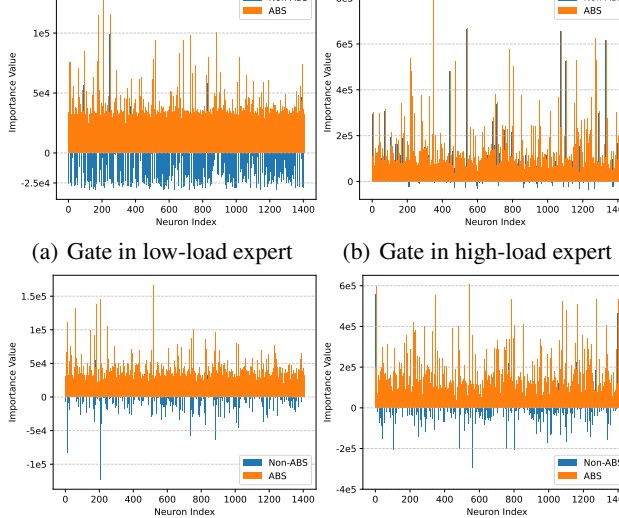

(c) Gate-Up in low-load expert  (d) Gate-Up in high-load expert

*Figure 9.* Comparison of neuron importance values derived from four profiling methods for expert 15 (high-load) and expert 21 (low-load) in layer 20 of DeepSeek-V2-Lite-Chat.

### 4.3.4. ANALYSIS OF NEURON IMPORTANCE PROFILING

2T-Drop with expert reconstruction is training-free, requiring only a lightweight recalibration stage to collect profiling statistics without updating model parameters. This recalibration involves two key design choices: the neuron importance profiling method and the calibration dataset. Notably, the overhead of recalibration under different configurations is minimal. For example, profiling requires only a single inference pass over the selected calibration dataset, followed by one verification pass on all downstream benchmarks; in our experiments, this process can be completed within minutes.

Following the introduction of four profiling methods in Section 3.3, we adopt the accumulated absolute gate value as the importance metric for Mixtral and OLMoE, and the accumulated absolute gate-up value for DeepSeek. Using DeepSeek-V2-Lite-Chat as an example, we observed the following average accuracies: accumulated gate (67.17%), accumulated absolute gate (67.29%), accumulated gate-up (66.79%), and accumulated absolute gate-up (67.65%).

Figure 9(a) and Figure 9(b) reveals that while low-load experts frequently exhibit negative accumulated gate values, high-load experts do not. This contrast underscores the interconnection between tensor- and neuron-level activations of MoE architectures. Conversely, accumulated gate-up values (Figure 9(c) and Figure 9(d)) show similar distributions across experts regardless of load, potentially accounting for this method's effectiveness for the DeepSeek model.

Regarding calibration samples, we evaluate different datasets as well as mixtures of varying sizes sampled from

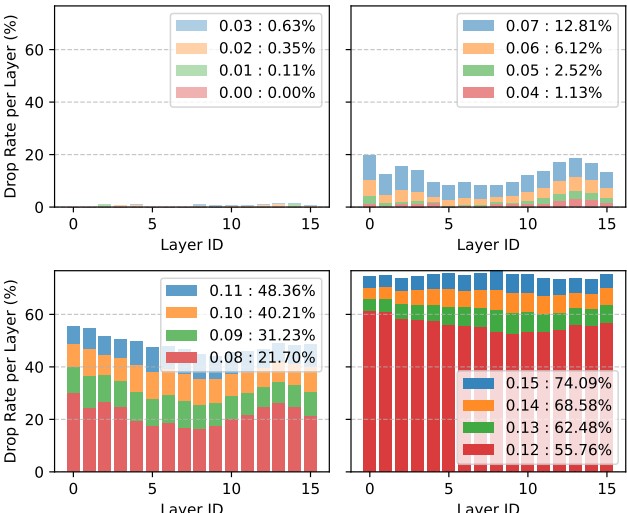

*Figure 10.* Drop rates across different layers of the OLMoE-Instruct model as a function of varying thresholds. The proportions in legend represent the overall drop rate of all layers.

the eight downstream tasks. The results show that different datasets lead to approximately 0.3% fluctuation in average accuracy. Accordingly, we use MMLU as the calibration dataset, since it demonstrates strong generalization across the evaluated models and downstream benchmarks.

In summary, neuron importance profiling methods and calibration datasets warrant further investigation across different models and deployment scenarios, and may be automatically and dynamically configured by the inference system in practical deployments.

### 4.3.5. ANALYSIS OF THRESHOLD AND DROP RATE

The threshold value directly controls the computation drop rate and thus governs the trade-off between accuracy and efficiency. We analyze this relationship across the evaluated models. Figure 10 shows the results for OLMoE-Instruct, where the drop rate changes nonlinearly as the threshold increases. This observation suggests that the threshold should be mapped to the target drop rate in a model-specific manner. In our evaluations, we empirically set the dropping threshold to obtain substantial speedup while limiting the average accuracy degradation to within 0.5%. In practical deployment, this threshold can be adjusted according to scenario-specific requirements and automatically configured by the system. Additionally, we observe that drop rates differ across layers, suggesting that per-layer thresholding may further improve the accuracy-efficiency trade-off in future work.

### 4.4. Advancements over Related Work

We compare our approach against existing sparsity-based methods: Efficient Expert Skipping (EES) (Lu et al., 2024) for dynamically bypasses expert computation for acceleration, and Efficient Expert Pruning (EEP) (Lu et al., 2024) for

*Table 4.* Comparison of our methods with EES and EEP (Lu et al., 2024) on Mixtral-8×7B-Instruct model inference.

| Method | Memory | Speedup | GSM8K Acc. Variation (↑) |
|---|---|---|---|
| 1T-Drop (Accuracy-Oriented) | - | 1.00× | +2.5% |
| 2T-Drop (Reconstruct) | - | 1.08× | +1.1% |
| EES | - | 1.05× | -2.4% |
| EEP ($r = 6$) | -24% | 1.20× | -8.0% |
| EEP ($r = 6$) + EES | -24% | 1.28× | -14.9% |

static pruning non-critical experts for compression. As both EES and our method target FLOP reduction, a comparative analysis reveals that our approach delivers superior accuracy (+1.1% vs. -2.4%) and greater speedup (1.08× vs. 1.05×).

While comparing dynamic computation reduction with static model compression is not strictly equivalent, it offers valuable insights. Notably, static expert pruning often incurs significant accuracy degradation compared to dynamic reduction, underscoring the necessity of maintaining dynamic activation capabilities. This degradation is also evident in weight pruning methods such as Wanda (Sun et al., 2024), which results in a severe GSM8K accuracy drop of 50.7% under a 2:4 sparsity pattern. In contrast, our proposed 2T-Drop (reconstruct) method integrates dynamic tensor-level dropping with static neuron-level weight differentiation to achieve enhanced performance.

Furthermore, our comparative experiments exclude load-aware thresholding, as prior studies have largely overlooked distributed MoE inference with EP and have not adequately addressed its inherent load imbalance. It is also crucial to distinguish between deployment scenarios: while edge environments prioritize model compression to fit limited device capacities, server-side deployments typically leverage distributed inference, where the primary objective is maximizing accuracy and throughput.

## 5. Conclusion

While fine-grained MoE architectures offer high tensor-level sparsity to achieve an excellent balance between accuracy and efficiency during pre-training, we observe that this sparsity is not fully exploited during post-training and inference. To address this, we introduce complete expert partition and threshold-based token-expert dropping to enhance accuracy during the post-training and inference stages, respectively. Furthermore, to optimize the accuracy-efficiency trade-off specifically for inference, we propose dual-threshold token-expert dropping combined with partial expert partition and reconstruction, leveraging sparsity at both the tensor and neuron levels. Experimental results substantiate the improvements in both accuracy and efficiency. Additionally, as discussed in Appendix C, we propose Soft Expert-Tensor Parallelism (S-ETP) upon expert partition, which enables tensor-level partitioning of expert weights via an algorithmic approach and facilitates higher communication bandwidth.

## Acknowledgements

We would like to thank the anonymous reviewers for their constructive feedback. This work was supported in part by the National Key R&D Program of China (No. 2024YFB4505800), the Guangdong Basic and Applied Basic Research Foundation (No. 2023A1515110353), and the Guangdong Provincial Project (No. 2023QN10X252).

## Impact Statement

This paper presents work whose goal is to advance the field of Machine Learning. There are many potential societal consequences of our work, none of which we feel must be specifically highlighted here.

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

## A. Expert Partition Details

### A.1. Complete Expert Partition for Post-Training

Complete transformation partitions each expert of a pre-trained MoE model into $P$ finer-grained experts (e.g. $P = 2$, as shown in Figure 2(b)). This approach allows the transformed MoE model to integrate seamlessly with existing MoE frameworks, functioning identically to the original model. Specifically, the transformation involves three steps: (1) Repeat the gating network weights $P$ times and adjust the Top-$K$ selection to Top-$(K \times P)$; (2) Evenly partition the original experts' neurons into $P$ finer-grained experts; (3) Scale the down-projection weight $\mathbf{W}_2$ of each partitioned expert by a factor of $P$.

Next, we provide a formal derivation to demonstrate that this transformation ensures mathematical consistency. According to Equation (1) in Section 2.1, MoE module first employs $\mathbf{W}_g = [h_1, h_2, \ldots, h_E] \in \mathbb{R}^{d_{model} \times E}$ to compute the gating logits $\mathbf{l}$, where each $h_i$ is a vector of dimension $d_{model}$. Given an input token $\mathbf{x}_i$, its gating logits are computed as:

$$\mathbf{l} = \mathbf{x}_i \cdot \mathbf{W}_g = [l_1, l_2, \ldots, l_E]. \tag{12}$$

These gating logits are passed through a softmax function to obtain the gating scores $\mathbf{s} = [s_1, s_2, \ldots, s_E]$, where the gating score $s_e$ for the expert $e$ is calculated as follows:

$$s_e = \frac{\exp(l_e)}{\sum_{i=1}^{E} \exp(l_i)}. \tag{13}$$

In the complete transformation, each vector $h_e$ in $\mathbf{W}_g$ is repeated $P$ times to construct the new gating weight matrix $\mathbf{W}_g^P \in \mathbb{R}^{d_{model} \times (E \times P)}$, defined as:

$$\begin{aligned} \mathbf{W}_g = [&h_{1,1}, h_{1,2}, \ldots, h_{1,P}, h_{2,1}, h_{2,2}, \ldots, h_{2,P}, \\ &\ldots, h_{E,1}, h_{E,2}, \ldots, h_{E,P}] \end{aligned} \tag{14}$$

where $h_{e,p}$ denotes the $p$-th copy of the $e$-th original expert-specific vector. Accordingly, the new gating logits $\mathbf{l}^P$ for an input token $\mathbf{x}_i$, obtained via $\mathbf{W}_g^P$, can be expressed as:

$$\begin{aligned} \mathbf{l}^P = \mathbf{x}_i \cdot \mathbf{W}_g^P = [&l_{1,1}, l_{1,2}, \ldots, l_{1,P}, \\ &l_{2,1}, l_{2,2}, \ldots, l_{2,P}, \ldots, l_{E,1}, l_{E,2}, \ldots, l_{E,P}], \end{aligned} \tag{15}$$

where $l_{e,1} = l_{e,2} = \ldots = l_{e,P}$ due to the repeated vectors $h_{e,1} = h_{e,2} = \ldots = h_{e,P}$ in $\mathbf{W}_g^P$. Given the extended gating logits $\mathbf{l}^P$, the gating score for each finer-grained expert $s_{e,p}$ is calculated as:

$$s_{e,p} = \frac{\exp(l_{e,p})}{\sum_{j=1}^{E} \sum_{k=1}^{P} \exp(l_{j,k})} = \frac{1}{P} \cdot \frac{\exp(l_e)}{\sum_{j=1}^{E} \exp(l_j)}. \tag{16}$$

Since all $P$ finer-grained experts partitioned from the same original expert share identical gating scores, they are activated together under the Top-$(K \times P)$ selection mechanism.

Moreover, the sum of their outputs equals the original expert output, as shown below:

$$f_e(\mathbf{x}_i) = \sum_{p=1}^{P} f_{e,p}(\mathbf{x}_i), \tag{17}$$

which is analogous to the effect of tensor parallelism. Consequently, the output $\mathbf{y}_i^P$ of the partitioned MoE module for an input token $\mathbf{x}_i$ can be formulated as:

$$\begin{aligned} \mathbf{y}_i^P &= \sum_{e=1}^{E} \sum_{p=1}^{P} \frac{1}{P} \cdot \frac{\exp(l_{e,p})}{\sum_{j=1}^{E} \sum_{k=1}^{P} \exp(l_{j,k})} \cdot f_{e,p}(\mathbf{x}_i) \\ &= \frac{1}{P} \sum_{e=1}^{E} \cdot \frac{\exp(l_e)}{\sum_{j=1}^{E} \exp(l_j)} \cdot \sum_{p=1}^{P} f_{e,p}(\mathbf{x}_i) = \frac{\mathbf{y}_i}{P}. \end{aligned} \tag{18}$$

This derivation shows that the complete transformation preserves the overall MoE output by a scaling factor of $P$.

To ensure that $\mathbf{y}_i^P$ is equivalent to the original output $\mathbf{y}_i$, the result must be scaled by a factor of $P$. There are two ways to achieve this: (1) multiplying the gating scores by $P$, or (2) scaling the expert weight $\mathbf{W}_2$ of the down-projection by $P$. To preserve the original model structure without modifying the existing framework, we choose to scale the expert weights for complete transformation. For example, in Figure 2(b), the down-projection weights $\mathbf{W}_2$ of each partitioned expert is multiplied by 2, as $P = 2$.

### A.2. Partial Expert Partition for Inference

Partial transformation offers an alternative approach for partitioning experts in pre-trained MoE models, as illustrated in Figure 2(c). In contrast to complete transformation, partial transformation preserves the original gating network and only modifies the results of the Top-$K$ selection through two operations: (1) repeating the gating scores and (2) remapping the expert indices. Specifically, the original gating scores of the selected expert $[s_1, s_2, \ldots, s_K]$ are repeated $P$ times, resulting in $[s_1, s_2, \ldots, s_K]^P$. The corresponding expert indices $\mathbf{I} = [i_1, i_2, \ldots, i_K]$ are remapped as follows:

$$\begin{aligned} \mathbf{I}^P = [&i_1 P, i_2 P, \ldots, i_K P, i_1 P + 1, i_2 P + 1, \ldots, i_K P + 1, \\ &\ldots, i_1 P + P - 1, i_2 P + P - 1, \ldots, i_K P + P - 1], \end{aligned} \tag{19}$$

where each original expert is partitioned and placed contiguously, maintaining its relative position. Each expert is evenly split into $P$ finer-grained experts without scaling the down-projection weight. This is because the product of the partitioned experts' outputs and the repeated gating scores reproduces the original MoE outputs, formulated as follows:

$$\mathbf{y}_i^P = \sum_{e=1}^{E} \cdot \frac{\exp(l_e)}{\sum_{j=1}^{E} \exp(l_j)} \cdot \sum_{p=1}^{P} f_{e,p}(\mathbf{x}_i) = \mathbf{y}_i. \tag{20}$$

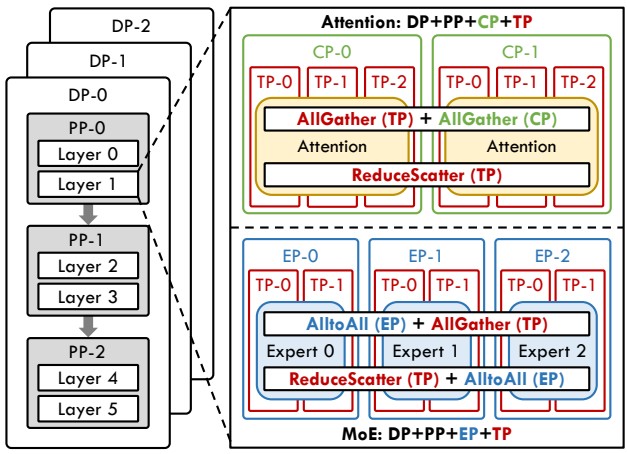

*Figure 11.* An example of a state-of-the-art 5-D hybrid parallel strategy for MoE model deployment. For simplicity, the illustration omits that EP can be extended cross DP groups. Our contributions mainly focus on improving the MoE part.

While partial transformation requires additional modifications to the existing MoE framework, it reduces computational overhead compared to the extended gating network required by complete transformation, despite the gating network constituting only a minor portion of the overall MoE layer's cost. Moreover, partial transformation maintains the original gating network parameters, enabling mathematically consistent reverse transformation and focusing solely on system efficiency. Consequently, we apply partial transformation to the Soft Expert-Tensor Parallelism introduced in Appendix C, as well as to the 2T-Drop inference described in Section 3.3.

## B. Hybrid Parallelism for MoE Model Deployment

Scaling the training and inference of MoE models across distributed devices requires an effective hybrid parallelism strategy. In this work, we adopt one of the state-of-the-art hybrid parallelism strategies (Nvidia, 2025a), as illustrated in Figure 11, to demonstrate the deployment pattern. This approach integrates Data Parallelism (DP) (Rajbhandari et al., 2020; Ren et al., 2021; Rajbhandari et al., 2021), Pipeline Parallelism (PP) (Huang et al., 2019; Narayanan et al., 2019), Expert Parallelism (EP) (Lepikhin et al., 2021; Fedus et al., 2022; Singh et al., 2023), Tensor Parallelism (TP) (Shoeybi et al., 2019; Smith et al., 2022; Narayanan et al., 2021), and Context Parallelism (CP) (Nvidia, 2025b; Korthikanti et al., 2023). Recent studies further suggest decoupling attention and MoE modules to enable more efficient resource allocation (Liu et al., 2025; Zhu et al., 2025). Although specific implementations differ—for instance, EP may be realized using either AlltoAll or AllGather—this 5-D parallelism strategy is broadly representative and offers

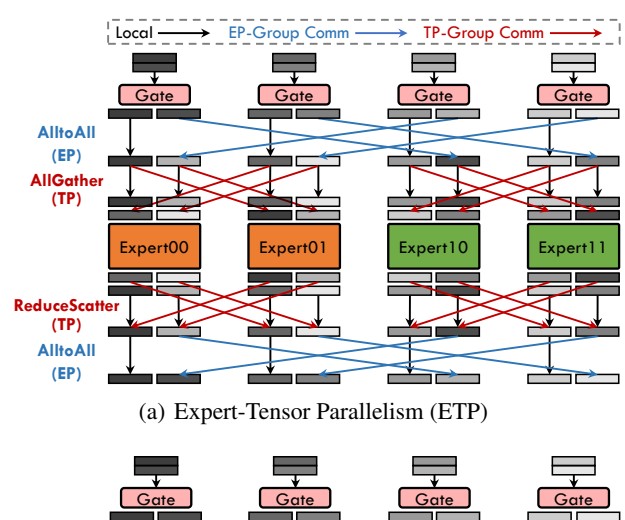

(a) Expert-Tensor Parallelism (ETP)

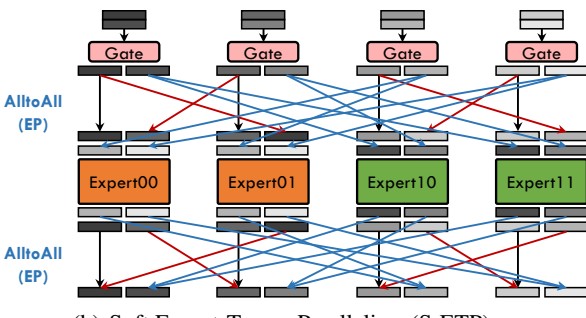

(b) Soft Expert-Tensor Parallelism (S-ETP)

*Figure 12.* Communication patterns in (a) Expert-Tensor Parallelism and (b) Soft Expert-Tensor Parallelism.

a general reference for both training and inference.

Importantly, tensor-level sparsity directly influences the choice and configuration of parallel strategies. In MoE layers, the number of experts and their intermediate sizes primarily affect EP and TP, since these parallel strategies handle the distribution of computational and memory loads and rely on communication patterns such as "AlltoAll+AllGather" and "ReduceScatter+AlltoAll." Therefore, the parallel strategy must comprehensively balance FLOPs utilization, communication overhead, GPU memory capacity, and other relevant factors.

## C. Soft Expert-Tensor Parallelism (S-ETP)

As discussed in Appendix B, EP and TP are employed to scale MoE deployment across distributed devices. In this context, applying TP to partition expert weights within EP is commonly referred to as Expert-Tensor Parallelism (ETP) (Liu et al., 2025; Singh et al., 2023).

In contrast, we propose Soft Expert-Tensor Parallelism (S-ETP), which enables tensor-level partitioning of expert weight through an algorithmic approach rather than relying solely on system-level implementation. Specifically, S-ETP integrates partial expert partition with EP to achieve the

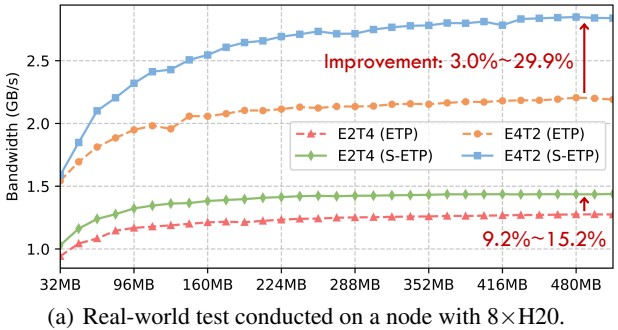

(a) Real-world test conducted on a node with 8×H20.

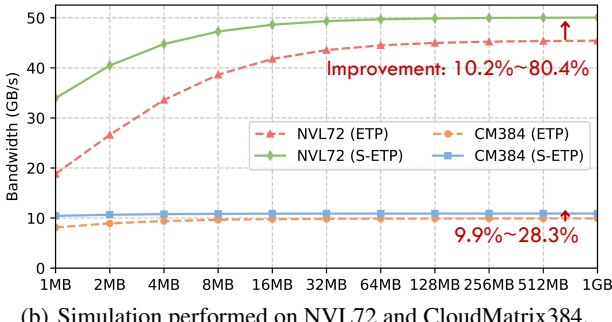

(b) Simulation performed on NVL72 and CloudMatrix384.

*Figure 13.* Comparison of communication bandwidth across different input sizes using ETP and S-ETP. In real-world tests (a), "E2T4" denotes a configuration with EP=2 and TP=4, while "E4T2" denotes a configuration with EP=4 and TP=2. In simulation (b), NVL72 (Nvidia, 2025c) is configured with EP=9 and TP=8, whereas CloudMatrix384 (CM384) (Zuo et al., 2025) is set to EP=48 and TP=8.

*Table 5.* Downstream performance comparison on two additional open-source MoE models.

| Model | Drop Method | HumanEval | MATH500 | GSM8K | ARC-C | BoolQ | HellaSwag | MMLU | OBQA | PIQA | RTE | WinoGrande | AVG.(↑) |
|---|---|---|---|---|---|---|---|---|---|---|---|---|---|
| Qwen3-30B-A3B | No Drop | 77.4 | 81.8 | 87.1 | 56.5 | 88.7 | 77.7 | 77.8 | 44.6 | 80.4 | 82.3 | 70.4 | 75.0 |
| | 2T (Reconstruct) | 81.1 | 80.7 | 87.9 | 54.9 | 88.4 | 76.8 | 77.2 | 43.0 | 81.1 | 83.0 | 70.0 | 74.9 |
| GPT-OSS-20B | No Drop | 73.8 | 75.4 | 84.3 | 48.7 | 76.1 | 58.0 | 56.6 | 40.4 | 77.4 | 69.5 | 66.1 | 66.0 |
| | 2T (Reconstruct) | 73.5 | 77.6 | 83.2 | 48.2 | 75.2 | 57.9 | 55.0 | 40.0 | 76.9 | 71.5 | 66.1 | 65.9 |

same functionality as ETP.

S-ETP offers the following advantages: **(1) Reduced Framework Complexity.** ETP often requires additional control mechanisms and framework modifications. In contrast, S-ETP addresses these challenges from an algorithmic rather than a system perspective, requiring only EP implementations, and thereby simplifying system optimization efforts. **(2) Optimized Communication Patterns.** S-ETP uses only the AlltoAll operation (Figure 12(b)) to achieve the same effects as the "AlltoAll+AllGather" and "ReduceScatter+AlltoAll" patterns used in ETP (Figure 12(a)). This approach reduces kernel launches and synchronization overhead, improving interconnect link utilization in both training and inference.

In addition to optimizing scenarios that traditionally require ETP, our expert partition approach also benefits cases that require scaling up EP. Specifically, our method enables the deployment of a larger number of experts, thereby involving more EP devices and enhancing scalability. Furthermore, the aforementioned advantages are also applicable to models restructured using complete expert transformation.

### C.1. Efficiency Improvements Achieved via S-ETP

We perform small-scale real-world tests using the PyTorch Distributed framework with the NCCL backend, as well as large-scale simulations using the ASTRA-SIM simulator (Rashidi et al., 2020) to evaluate the optimization achieved by the Soft Expert-Tensor Parallelism (S-ETP).

In Figure 13, S-ETP exhibits significant improvements in communication bandwidth compared to existing ETP approach. The bandwidth is measured by dividing the input size per device by the total communication time. In a real-world test configuration of EP=4 and TP=2 on an 8×H20 node, S-ETP achieves a bandwidth improvement ranging from 3.0% to 29.9%. When configured with EP=2 and TP=4, the improvement ranges from 9.2% to 15.2%.

Furthermore, the benefits of S-ETP are particularly evident in systems equipped with fully peer-to-peer high-bandwidth interconnections, such as NVL72 (Nvidia, 2025c) and CloudMatrix384 (Zuo et al., 2025). These systems feature homogeneous network architectures, eliminating the substantial disparities typically observed between inter-node and intra-node bandwidth. Our simulations in these environments reveal improvements of 10.2% to 80.4% on NVL72 and 9.9% to 28.3% on CloudMatrix384.

## D. Evaluation on Additional MoE Models

We further evaluate the proposed dropping strategy on two additional open-source MoE models, Qwen3-30B-A3B (Yang et al., 2025) and GPT-OSS-20B (Agarwal et al., 2025). For the 2T-Drop (Reconstruct) strategy, we use $T_{major}^2 = 0.07$ and $T_{minor}^2 = 0.09$ for Qwen3-30B-A3B, and $T_{major}^2 = 0.14$ and $T_{minor}^2 = 0.16$ for GPT-OSS-20B. As shown in Table 5, the dropping method achieves comparable average accuracy to the no-drop baseline across a broad range of benchmarks.

