# OpenReview forum: "Mining Tensor/Neuron-Level Sparsity to Maximize Mixture-of-Experts Potential in Post-Training and Inference"
_ICML.cc/2026/Conference — ICML 2026 regular_

### Official Review · Reviewer_QJNf · 2026-02-22

**Soundness:** 3
**Presentation:** 3
**Significance:** 3
**Originality:** 3
**Overall Recommendation:** 4
**Confidence:** 3

**Summary:**

This paper identifies underexploited dual sparsity in MoE models at both tensor level (which experts are activated) and neuron level (which neurons within experts contribute significantly). For post-training, it proposes complete expert partition, which splits each expert into finer-grained sub-experts while maintaining mathematical equivalence, yielding lower SFT loss and higher downstream accuracy. For inference, it introduces threshold-based token-expert dropping (1T-Drop), which filters low-gate-score computations as noise to improve accuracy. To further optimize the accuracy-efficiency trade-off, it proposes 2T-Drop, combining partial expert partition with neuron importance-based reconstruction into major/minor sub-experts and applying differentiated thresholds, along with load-aware thresholding for expert-parallel deployment.

**Compliance With Llm Reviewing Policy:**

Affirmed.

**Final Justification:**

After further discusstion, the rebuttal of authors have addresses most of my concerns, so I raised my score from 3 (weak reject) to 4 (weak accept).

**Key Questions For Authors:**

1. Regarding the actual inference speedup, I note that popular inference frameworks such as vLLM typically adopt a fused MoE design for the MLP portion, where different experts are combined into a single large sparse matrix for joint computation. In my experience, under this fused setting, expert pruning and parameter sparsity often fail to achieve meaningful wall-clock speedup, and load imbalance across experts does not bottleneck overall computation either. Could the authors discuss whether and how this fused MoE execution pattern affects the practical acceleration benefits of the proposed methods?
  2. Referring to the weaknesses above, if the authors add the discussion and theoretical analysis mentioned in W1 and supplement experiments on at least one newer MoE architecture as suggested in W2, I am willing to raise my score to 4 or 5.

**Limitations:**

yes

**Strengths And Weaknesses:**

Strengths:
  1. The paper is comprehensive in its exploration of sparse inference acceleration for MoE models. While some claims could be better substantiated, the depth and breadth of the discussion — covering tensor-level and neuron-level sparsity, post-training and inference stages, as well as distributed deployment considerations — demonstrate a thorough understanding of the MoE inference landscape.
  2. The idea of jointly leveraging expert-level pruning/dropping and neuron-level sparsity within a unified framework is novel and, in my view, represents a promising research direction. From this perspective, the paper is inspiring for future work on MoE efficiency optimization.
  3. The experimental evaluation is relatively comprehensive, covering both zero-shot classification tasks and reasoning benchmarks, as well as actual wall-clock inference speedup measurements. Although the reasoning evaluation is limited to a single dataset, I still view it as a strength given that many comparable works omit reasoning tasks entirely.

Weaknesses:
  1. The claims and methodology in the "Complete Expert Partition for Post-Training" section lack sufficient justification. Based on recent literature and my own experimental observations, fine-grained MoE models (e.g., Qwen3-MoE, Qwen3-Next, GPT-OSS, DeepSeek-V2/V2.5/V3) — which employ more experts per layer and activate more experts per token — inherently exhibit higher expert redundancy (i.e., tensor-level sparsity), whereas coarse-grained MoE models like Mixtral-8×7B with only top-2 routing have relatively lower redundancy. **Meanwhile, there is a broad consensus in both academia and industry that fine-grained MoE architectures are superior to coarse-grained ones in terms of training performance. I suspect this is the fundamental reason why the proposed expert partition improves fine-tuning outcomes** — it essentially transforms a coarse-grained MoE into a fine-grained one. However, the paper does not discuss the well-established coarse-to-fine-grained MoE evolution and provides no theoretical analysis of why partitioning experts from a coarse-grained MoE raises the fine-tuning performance ceiling. This omission weakens the contribution of this component.
  2. The paper acknowledges that the neuron importance profiling method must be empirically selected per model, lacking a universal selection criterion. To compensate for this limitation, I believe the authors should validate their approach on more recent MoE models. All models evaluated in the paper were released in 2024, and there is no verification on newer architectures such as Qwen3-MoE, Ling-Lite-MoE, or GPT-OSS, which would help assess the generalizability of the proposed methods and profiling strategies.

---

> ### Author Rebuttal · Authors · 2026-03-31
>
> > Q1: The claims and methodology in the "Complete Expert Partition for Post-Training" section lack sufficient justification. The paper does not discuss the well-established coarse-to-fine-grained MoE evolution and provides no theoretical analysis of why partitioning experts from a coarse-grained MoE raises the fine-tuning performance ceiling.
>
> Thank you for your valuable feedback. Our exploitation of the coarse-to-fine-grained MoE transformation during post-training and inference is exactly inspired by the widely recognized trend of fine-grained expert design in pre-training. We explicitly discussed this motivation in the second paragraph of the Introduction section: *"During pre-training, existing research pursues higher tensor-level sparsity through finer-grained expert designs ... As deployment conditions and workloads differ significantly from the pre-training, static expert configurations established during pre-training fail to capture additional sparsity opportunities available during these subsequent stages."*
>
> To provide a deeper theoretical understanding of why expert partitioning improves the fine-tuning performance ceiling, we supplement the following analysis, building upon our formulation in Appendix A.1.
> Fundamentally, partitioning an original expert into $P$ finer-grained experts increases the learnable parameters of the MoE router by a factor of $P$. Although this marginal parameter increase brings no noticeable computational overhead, it substantially enhances the router's classification capacity. Since the core advantage of an MoE architecture over a dense MLP lies in distilling inherent token-specific activation patterns into explicit routing classifications, this enhancement yields a superior accuracy-efficiency trade-off that is highly beneficial for both pre-training and post-training.
>
> Unlike pre-training from scratch with random initialization, our method leverages a mathematically consistent transformation to repeat pre-trained router embeddings for further tuning. This structurally benefits the router's classification learning and the experts' downstream regression during post-training, thereby effectively raising the fine-tuning performance ceiling.
>
> > Q2: The paper acknowledges that the neuron importance profiling method must be empirically selected per model, lacking a universal selection criterion. To compensate for this limitation, I believe the authors should validate their approach on more recent MoE models.
>
> To address concerns regarding the generalizability of our methods, we provide supplementary evaluations on the Qwen3-30B-A3B and GPT-OSS-20B models, including two additional benchmarks, HumanEval and MATH500, for a more comprehensive evaluation.
> As shown in the following table, our methods consistently achieve efficiency improvements without compromising accuracy.
> |Model|Method|HumanEval|MATH500|GSM8K|ARC-C|BoolQ|HellaSwag|MMLU|OBQA|PIQA|RTE|WinoGrande|AVG|MoE Speedup|
> |-|-|-|-|-|-|-|-|-|-|-|-|-|-|-|
> |Qwen3-30B-A3B|No Drop|32.3|57.0|89.7|56.5|88.7|77.7|77.8|44.6|80.4|82.3|70.4|68.9|1|
> |Qwen3-30B-A3B|2T-Drop (Reconstruct)|35.9|57.2|88.7|54.9|88.4|76.8|77.2|43.0|81.1|83.0|70.0|68.7|1.16|
> |GPT-OSS-20B|No Drop|28.6|11.6|36.7|48.7|76.1|58.0|56.6|40.4|77.4|69.5|66.1|51.8|1|
> |GPT-OSS-20B|2T-Drop (Reconstruct)|30.5|13.4|36.5|48.2|75.2|57.9|55.0|40.0|76.9|71.5|66.1|51.9|1.21|
>
> > Q3: Could the authors discuss whether and how this fused MoE execution pattern affects the practical acceleration benefits of the proposed methods?
>
> The practical benefit of expert pruning depends on the operating regime:
> 1. **Compute-bound regime (e.g., multi-sequence prefilling).** When token volume is large, fused kernel can saturate GPU compute resources. In this case, pruning-induced computation reduction can lead to direct acceleration.
> 2. **Memory-bound regime (e.g., decoding with few active tokens).** When only a few tokens are active, latency is dominated by memory accesses to expert weights. Expert pruning can still accelerate by reducing the number of activated experts and the corresponding memory traffic.
> 3. **Intermediate regime.** In between, the realized speedup is generally smaller than the nominal FLOPs reduction, and practical gains may require sufficiently high sparsity.
> Moreover, the overhead of the pruning method itself must remain small; otherwise, the benefit from reduced computation may be diminished.
>
> More broadly, expert pruning is typically more hardware-friendly because it removes computation in a coarse-grained and structured manner, but it often causes larger accuracy loss. Parameter sparsity is usually more accuracy-preserving, yet its fine-grained irregularity makes efficient execution more difficult in practice.
>
> By leveraging dual sparsity, our method combines the hardware efficiency of expert-level pruning with the accuracy benefits of neuron-level sparsification, achieving a better balance between practical acceleration and model quality.

---

> > ### Author Rebuttal · Reviewer_QJNf · 2026-04-02
> >
> > Thank you for the detailed explanation and the additional experiments. However, in the rebuttal I noticed a clear issue:
> >
> > The reproduced results for Qwen3-30B-A3B on HumanEval and MATH500, and for GPT-OSS-20B on HumanEval, MATH500, and GSM8K, appear to be **substantially underestimated**. For example, on MATH500, both Qwen3-30B-A3B and GPT-OSS-20B should achieve at least over 90% accuracy. Even conservatively speaking, MATH500 contains around one hundred Level-1 middle-school-level math problems; if GPT-OSS-20B really achieves only 11.6% accuracy, then it would perform worse than I do myself. I am quite familiar with the performance of these fine-grained MoE models on mathematical and code reasoning tasks, so I am fairly confident that these results are incorrect. The authors may also consult the technical reports of these two models, where it can be seen that both models achieve over 70% accuracy even on competition-level benchmarks such as AIME25. Alternatively, the authors may refer to the [Math-500 leaderboard](https://artificialanalysis.ai/evaluations/math-500) to verify the approximate baseline range for models at a similar capability level.
> >
> > Although the reproduction appears clearly flawed, for now I am inclined to interpret this as a consequence of unfamiliarity with evaluation for reasoning tasks, rather than intentional misrepresentation. I would encourage the authors to carefully check the evaluation toolkit they used—for example, platforms such as OpenCompass or LightEval may be more appropriate for reasoning tasks—and also verify whether the maximum generation length is set properly (for these two models, it should likely be at least 16k). I am willing to remain patient and wait for the authors to provide corrected and reasonable results, and I will make my final judgment based on that.

---

> > > ### Author Response · Authors · 2026-04-02
> > >
> > > Thank you very much for your careful comment and guidance on this issue.
> > >
> > > In the previous rebuttal, we evaluated all 11 tasks jointly using lm-evaluation-harness with its default configuration, including the default sequence length settings. We agree that this setup was not suitable for reasoning benchmarks, which resulted in substantially underestimated scores for Qwen3-30B-A3B and GPT-OSS-20B.
> > >
> > > To address this issue, we have re-run the reasoning-task evaluation using `OpenCompass`, with the maximum generation length increased to `16K`. The updated results are reported below. As can be seen, the reproduced scores are significantly higher than those in our earlier rebuttal and are much more aligned with the expected capability range of these models.
> > >
> > > At the same time, our method continues to achieve comparable accuracy relative to the No-Drop baseline under the corrected evaluation protocol.
> > >
> > > | Model | Method | HumanEval | MATH500 | GSM8K |
> > > | - | - | -: | -: | -: |
> > > | Qwen3-30B-A3B | No Drop | 77.4 | 81.8 | 87.1 |
> > > | Qwen3-30B-A3B | 2T-Drop (Reconstruct) | 81.1 | 80.7 | 87.9 |
> > > | GPT-OSS-20B | No Drop | 73.8 | 75.4 | 84.3 |
> > > | GPT-OSS-20B | 2T-Drop (Reconstruct) | 73.5 | 77.6 | 83.2 |
> > >
> > > We sincerely appreciate the reviewer’s careful comment, which helped us identify this evaluation issue and improve the reliability of our reported results. If you have any additional questions or comments, we would be happy to discuss them further.

---

### Official Review · Reviewer_DyfF · 2026-03-03

**Soundness:** 3
**Presentation:** 4
**Significance:** 3
**Originality:** 3
**Overall Recommendation:** 5
**Confidence:** 5

**Summary:**

This paper exploits underutilized tensor-level and neuron-level sparsity in MoE models during post-training and inference. It introduces complete expert partitioning to enhance fine-tuning performance, alongside a dual-threshold dropping strategy and neuron-level reconstruction to optimize the accuracy-efficiency trade-off.
Experimental results across multiple MoE models demonstrate that the proposed methods achieve significant speedup with minimal accuracy degradation.

**Compliance With Llm Reviewing Policy:**

Affirmed.

**Final Justification:**

A good paper and rebuttal solved most of my concerns so I decide to accept it.

**Key Questions For Authors:**

1. How can the thresholds for 1T-Drop and 2T-Drop be determined for a specific model?
2. Does the neuron-level importance remain consistent across different tasks? For instance, if you profile the model on MMLU, does the 2T-Drop strategy still maintain accuracy on other tasks?
3. How to choose the calibration dataset for neuron importance profiling?

**Limitations:**

Yes

**Strengths And Weaknesses:**

Strengths:
1. This paper moves beyond exsiting expert-level computation dropping by exploring the synergy between tensor-level and neuron-level sparsity. The idea of reconstructing experts based on neuron importance is novel.
2. The detailed analysis of sparsity patterns under real workloads provides valuable insights into expert activation behaviors, contributing to a deeper understanding of MoE dynamics.
3. The method is validated on multiple open-source models (e.g. Mixtral, DeepSeek, OLMoE) across diverse downstream benchmarks, demonstrating its practical applicability and generalizability.



Weaknesses:
1. The effectiveness of 1T-Drop and 2T-Drop depends on thresholds. The paper lacks a discussion on a systematic or automated way to determine these thresholds for new, unseen models.
2. The robustness of neuron reconstruction is potentially limited by its reliance on specific calibration datasets.

---

> ### Author Rebuttal · Authors · 2026-03-31
>
> > Q1: The effectiveness of 1T-Drop and 2T-Drop depends on thresholds. The paper lacks a discussion on a systematic or automated way to determine these thresholds for new, unseen models. How can the thresholds for 1T-Drop and 2T-Drop be determined for a specific model?
>
> Our approach involves empirically setting the dropping threshold to the maximum possible value that restricts the average accuracy degradation to approximately 0.5%. This strategy guarantees meaningful inference acceleration while strictly preserving the model's service quality.
>
> Because the overhead of running evaluation passes on downstream tasks is marginal (taking only minutes to complete), practitioners can comfortably evaluate several threshold candidates to identify the optimal configuration for any new model. Furthermore, this threshold serves as a flexible tunable parameter, allowing users to dynamically adjust it to satisfy the specific accuracy-efficiency trade-off requirements of different deployment scenarios.
>
> > Q2: The robustness of neuron reconstruction is potentially limited by its reliance on specific calibration datasets. How to choose the calibration dataset for neuron importance profiling?
>
> To address the concern regarding calibration data, we supplement our evaluation across different datasets and mixtures of various sizes sampled from the eight downstream tasks. As shown in the table below, compared to 1T-Drop (without expert reconstruction), 2T-Drop (with expert reconstruction) consistently improves the average accuracy across all evaluated tasks, regardless of the calibration dataset used, though the margin of improvement varies.
>
> We find that utilizing MMLU yields robust performance across all three tested models, and thus we adopt it as our default calibration dataset.
>
> In practice, practitioners can easily evaluate several candidate datasets and select the optimal one with negligible overhead. As discussed in Q4 of Reviewer Z2HT, this training-free calibration process only takes minutes to complete.
>
> | Calibration dataset | Avg. accuracy improvement on 8 tasks over 1T-Drop |
> |---|---|
> | MMLU | +0.38 |
> | Winogrande | +0.24 |
> | ARC-C | +0.10 |
> | Mix_sample_10K | +0.05 |
> | Mix_sample_20K | +0.33 |
>
> > Q3: Does the neuron-level importance remain consistent across different tasks? For instance, if you profile the model on MMLU, does the 2T-Drop strategy still maintain accuracy on other tasks?
>
> Thank you for raising this question. In the  manuscript, we have demonstrated that 2T-Drop (which performs expert reconstruction using neuron profiling evaluated on MMLU) maintains high accuracy across a diverse set of downstream tasks, ranging from multiple-choice question answering (e.g., MMLU) to mathematical reasoning (e.g., GSM8K).
>
> To further substantiate the cross-task consistency of our neuron-level profiling, we have expanded our evaluations to include more challenging generation and long-context benchmarks: HumanEval (code generation, https://arxiv.org/abs/2107.03374), MATH500 (mathematical reasoning, https://arxiv.org/abs/2305.20050), and RULER (comprehensive long-context evaluation, https://arxiv.org/pdf/2404.06654). RULER comprises four task categories (13 tasks in total) and evaluates model performance at context lengths up to 16K. All supplementary experiments strictly adhere to the configurations used in Table 2.
>
> As presented in the table below, our dropping method yields performance highly comparable to the no-drop baseline on both generation and long-context tasks. These robust results confirm that the neuron-level importance derived from MMLU calibration generalizes effectively and remains highly consistent across a wide spectrum of tasks.
>
> For OLMoE, because its max_position_embeddings is limited to 4K, evaluations beyond 4K lead to accuracy collapse and are therefore not meaningful as references. Hence, we omit the 8K and 16K results for OLMoE.
>
> | Model | Method | HumanEval | MATH500 | RULER-4K | RULER-8K | RULER-16K |
> |---|---|---|---|---|---|---|
> | DeepSeek-V2-Lite-Chat | No Drop | 48.2 | 22.4 | 75.7 | 84.4 | 81.6 |
> | DeepSeek-V2-Lite-Chat | 2T-Drop (Reconstruct) | 47.7 | 22.1 | 78.8 | 83.6 | 81.0 |
> | OLMoE-Instruct | No Drop | 34.8 | 19.4 | 61.9 | - | - |
> | OLMoE-Instruct | 2T-Drop (Reconstruct) | 35.4 | 19.8 | 61.7 | - | - |

---

> > ### Author Rebuttal · Reviewer_DyfF · 2026-04-01
> >
> > The author has effectively addressed most of my concerns. This is a solid and interesting paper, so I have decided to maintain my original score.
> >
> > Here is a small suggestion: Although the computational cost of a single threshold evaluation is relatively low, frequent testing can be time-consuming if there are many candidate hyperparameters to eliminate. I recommend adding an automated, controllable hyperparameter selection pipeline to the code.

---

> > > ### Author Response · Authors · 2026-04-01
> > >
> > > Thank you for your positive feedback. We are pleased to know that our rebuttal has addressed most of your concerns, and we appreciate your recognition that this is a solid and interesting paper.
> > >
> > > We also thank you for the valuable suggestion on hyperparameter selection. This is an important point from an system implementation perspective. We will further discuss this issue in the manuscript and add an automated, controllable hyperparameter selection pipeline to our project to make the system more practical and user-friendly.
> > >
> > > If you have any additional questions or comments, we would be happy to discuss them further.

---

### Official Review · Reviewer_etyj · 2026-03-04

**Soundness:** 3
**Presentation:** 2
**Significance:** 2
**Originality:** 2
**Overall Recommendation:** 4
**Confidence:** 2

**Summary:**

This paper argues that pretrained MoE LLMs leave exploitable sparsity at both the expert level and the neuron level during post-training and inference. It introduces complete expert partition, which splits each expert into finer sub-experts while keeping the MoE layer equivalent at initialization, and reports better fine-tuning loss and downstream accuracy on Mixtral-8×7B. It proposes threshold-based token-expert dropping (1T-Drop) and a dual-threshold scheme (2T-Drop) that reconstructs each expert into major/minor neuron groups and adds load-aware thresholds under expert parallelism. Experiments show modest accuracy gains and up to 1.4× MoE-layer speedups with minimal average accuracy loss.

**Compliance With Llm Reviewing Policy:**

Affirmed.

**Final Justification:**

Based on the current draft and rebuttal, I maintain my overall assessment as a weak accept. The paper is well motivated: it highlights a practical mismatch between fixed expert granularity from MoE pretraining and the needs of post-training/inference, and proposes a fairly complete engineering pipeline (complete expert partition plus 1T/2T drop) with clear derivations for the initialization-preserving partition. That said, the end-to-end gains are modest (roughly 1.07–1.13×), which makes the real-world payoff versus added system complexity less convincing, and the robustness of threshold/calibration choices across prompts, decoding settings, and long-context regimes would benefit from stronger evidence. The manuscript would also benefit from careful proofreading and refinement, but overall I think the ideas are sound and useful enough that I would not mind if the paper is accepted.

**Key Questions For Authors:**

1. For 1T-Drop and 2T-Drop, do you renormalize the remaining expert weights after dropping?

2. How sensitive are the accuracy and speed gains to the choice of calibration data and thresholds (e.g., different calibration sets, prompts, or decoding settings)?

3. Can you provide a breakdown of end-to-end latency/throughput by components (attention vs MoE vs others)?

**Limitations:**

The paper should discuss that threshold-based dropping may be brittle across prompts, domains, and long-context settings, and could degrade reliability on harder reasoning tasks.

**Strengths And Weaknesses:**

## Strengths
1. The motivation is clear. The paper explains the mismatch between fixed expert granularity/routing in pretraining and changing workloads at post-training/inference time. Fig. 1 is intuitive and helps.
2. The engineering story is fairly complete. The paper covers post-training, inference-time dropping, and practical issues under expert parallelism.
3. The method is supported by reasonably detailed derivations. The “equivalent initialization” for partitioning is explained clearly, which makes the proposal easier to trust and implement.

## Weaknesses
1. The system gets more complex, but the end-to-end speedup is only around 1.07–1.13×. It is unclear if the practical benefit justifies the added machinery.
2. Some key claims need stronger validation. Inference dropping is calibrated and tuned, but the paper does not fully establish robustness across different prompts, decoding settings, and context lengths, and the calibration choice may raise concerns about evaluation bias.

---

> ### Author Rebuttal · Authors · 2026-03-31
>
> > Q1: The system gets more complex, but the end-to-end speedup is only around 1.07–1.13×. It is unclear if the practical benefit justifies the added machinery.
>
> We believe the improvement is still practically meaningful for two reasons:
> 1. Our method is implemented on top of SGLang and is compatible with other optimization techniques, so its benefit is not isolated and can further improve existing systems, while having negligible impact on service quality.
> 2. In industrial serving environments, even a 10% throughput improvement is highly valuable, which can translate into 10% higher serving capacity and revenue impact.
>
> > Q2: The paper should discuss that threshold-based dropping may be brittle across prompts, domains, and long-context settings, and could degrade reliability on harder reasoning tasks.
>
> Thank you for raising this important concern. In the manuscript, we already evaluated our method across diverse settings and tasks, including multiple-choice question answering (e.g., MMLU) and mathematical reasoning (e.g., GSM8K).
>
> To further address the potential brittleness of threshold-based dropping across prompts, domains, and long-context scenarios, we additionally include evaluations on more challenging generation and long-context benchmarks: HumanEval (code generation), MATH500 (mathematical reasoning), and RULER (long-context evaluation). RULER (https://arxiv.org/pdf/2404.06654) comprises four task categories (13 tasks in total) and evaluates model performance at context lengths of 4K, 8K, and 16K. All experiments are conducted using the same configuration as in Table 2.
>
> As shown in the table below, our dropping method achieves performance comparable to the no-drop baseline on both generation and long-context tasks, suggesting that it does not introduce noticeable reliability degradation in these more challenging settings.
>
> For OLMoE, because its max_position_embeddings is limited to 4K, evaluations beyond 4K lead to accuracy collapse and are therefore not meaningful as references. Hence, we omit the 8K and 16K results for OLMoE.
>
> |Model|Method|HumanEval|MATH500|RULER-4K|RULER-8K|RULER-16K|
> |-|-|-|-|-|-|-|
> |DeepSeek-V2-Lite-Chat|NoDrop|48.2|22.4|75.7|84.4|81.6|
> |DeepSeek-V2-Lite-Chat|2T-Drop(Reconstruct)|47.7|22.1|78.8|83.6|81.0|
> |OLMoE-Instruct|NoDrop|34.8|19.4|61.9|-|-|
> |OLMoE-Instruct|2T-Drop(Reconstruct)|35.4|19.8|61.7|-|-|
>
> > Q3: How sensitive are the accuracy and speed gains to the choice of calibration data and thresholds?
>
> **Threshold Selection**: We empirically set the dropping threshold as high as possible while maintaining the average accuracy degradation within approximately 0.5%. This strategy achieves meaningful inference speedup without compromising the overall model service quality. The sensitivity analysis is provided in Figure 8, which illustrates that increasing the threshold yields higher speedups but inevitably leads to greater accuracy degradation.
>
> **Calibration Data Selection**: To address the concern regarding calibration data, we supplement our evaluation across different datasets and mixtures of various sizes sampled from the eight downstream tasks. As shown in the table below, compared to 1T-Drop (without expert reconstruction), 2T-Drop (with expert reconstruction) consistently improves the average accuracy across all evaluated tasks, regardless of the calibration dataset used, though the margin of improvement varies. We find that utilizing MMLU yields robust performance across all three tested models, and thus we adopt it as our default calibration dataset.
>
> In practice, practitioners can easily evaluate several candidate datasets and select the optimal one with negligible overhead. As discussed in [ Q4 of Reviewer Z2HT ], this training-free calibration process only takes minutes to complete.
>
> |Calibration dataset|Avg. accuracy improvement on 8 tasks over 1T-Drop|
> |-|-|
> |MMLU|+0.38|
> |Winogrande|+0.24|
> |ARC-C|+0.10|
> |Mix_sample_10K|+0.05|
> |Mix_sample_20K|+0.33|
>
>
> > Q4: For 1T-Drop and 2T-Drop, do you renormalize the remaining expert weights after dropping?
>
> No, we do not renormalize the remaining expert weights after dropping.
>
>
> > Q5: Can you provide a breakdown of end-to-end latency/throughput by components (attention vs MoE vs others)?
>
> We provide a breakdown of end-to-end latency to clarify the source of our speedup. Since our proposed methods exclusively target the optimization of the MoE modules, we present the relative latency contributions of the MoE, attention, and other components in the table below (and additionally visualize both the MoE-specific and end-to-end speedups in Figure 7 of the manuscript).
> The MoE module accounts for a significant portion (roughly half) of the total latency, demonstrating that optimizing this component translates to substantial end-to-end acceleration.
>
> | Model | MoE | Attention | Others |
> |---|---|---|---|
> | DeepSeek | 54% | 42% | 4% |
> | OLMoE | 45% | 52% | 3% |

---

> > ### Author Rebuttal · Reviewer_etyj · 2026-04-02
> >
> > I would like to thank the authors for the detailed rebuttal. I maintain my overall assessment as a weak accept.

---

> > > ### Author Response · Authors · 2026-04-02
> > >
> > > Thank you for your positive feedback.
> > > If you have any additional questions or comments, we would be happy to discuss them further.

---

### Official Review · Reviewer_Z2HT · 2026-03-12

**Soundness:** 3
**Presentation:** 1
**Significance:** 2
**Originality:** 3
**Overall Recommendation:** 4
**Confidence:** 4

**Summary:**

The paper says that pretrained MoE leaves useful sparsity unexploited during post-training and inference. It proposes complete expert partition, which splits each pretrained expert into finer-grained experts while preserving the original function at initialization, then fine-tunes the transformed model. For inference, they introduces threshold-based token-expert dropping, where routed experts with very small normalized gating scores are skipped. They further extend this to a dual-threshold method that splits each expert into major and minor sub-experts, so borderline experts can be executed only partially instead of fully kept or fully dropped. Experiments on various new MoE models show improvements in downstream accuracy and favorable inference speedup-accuracy tradeoffs.

**Compliance With Llm Reviewing Policy:**

Affirmed.

**Final Justification:**

The authors have provided thoughtful and helpful comments regarding my concerns. The proposed method does not require too much overhead, and the cost can be justified and amortized over time as it provides a better accuracy improvement on multiple tasks. While the paper presentation still needs more refinement for readability, I decided that this method needs more recognition, and thereby I increased the score for this paper.

**Key Questions For Authors:**

- What's the cost of finetuning? (# tokens, training steps) And what's the cost of it compared to prior works?
- The neuron importance profiling requires choosing both a metric and a calibration dataset, and the best metric differs across models. How sensitive are the 2T-Drop results to these choices? Specifically, what happens if you calibrate on a dataset from a very different distribution (e.g., code, long-form text), and what is the accuracy variance across the four profiling methods for each model? If the gap between the best and worst profiling metric is large, the method has a hidden tuning cost that should be discussed.
- All evaluations use short-context classification benchmarks. Have you tested whether 2T-Drop degrades generation quality on non-MCQ tasks that are modern and relevant?

**Limitations:**

They have not discussed the limitations in the paper. (There is no limitation section at all)

**Strengths And Weaknesses:**

**Strengths**
The paper has a clear and practical thesis: pretrained MoE models leave useful sparsity unexploited after pretraining, and that sparsity can be mined during post-training and inference. Also, their method does not degrade the performance, but rather increase the performance. The strongest aspect of this paper is the systems-aware design throughout. Many sparsity papers report theoretical FLOP reductions that never materialize as wall-clock gains. but this paper actually implements custom Triton kernels, measures real throughput on GPUs, and demonstrates that nearly all of the computation drop rate translates into proportional speedup. That dropping token-expert computations with very low normalized gating scores can actually improve accuracy is interesting and would be a useful finding for the community. It's also good that it's a post-training method that community can make use of (not only the frontier labs.)

**Weaknesses**
- The paper is a bit hard to read. Complete partition, partial partition, 1T-Drop, 2T-Drop, four neuron profiling methods, load-aware thresholding, and S-ETP are all introduced with their own notation and setup. One paper is trying to introduce too many ideas and the ideas are scattered throughout the paper.
- Weak related works in neuron-level sparsity. While the paper argues that "Most methods focus primiiarly on the ReLU activation function, ..., and therefore canndot be directly applied to modern LLMs employing SwiGLU activations". However, many papers like *Training-Free Activation Sparsity in Large Language Models* or *CATS: Contextually-Aware Thresholding for Sparsity in Large Language Models* exactly tackles this issue for SwiGLU. Addressing those papers and their relation to the proposed paper is lacking. They use the same thresholding techniques.
- All evaluations are on short-context MCQ benchmarks. It's not clear whether their methods can generalize to more important, relevant modern tasks (agent, long-horizon, math, etc.)
- It's not clearly how much additional overhead the finetuning costs. The papers mentioned above are almost training-free. It's skeptical how much finetuning steps and costs this method requires to recalibrate the model.
- The originality is limited. It's mostly a combination of already introduced techniques for pruning, newly applied in MoE.
- Also, it's not unsure the originality of their method. It seems like an incremental contribution that builds upon prior works that already do
- Grammar issues (*"output of MoE module is governed by **dual sparse** at both the tensor and neuron levels"* in pg1)

---

> ### Author Rebuttal · Authors · 2026-03-31
>
> > Q1: The paper is hard to read with too many ideas scattered throughout.
>
> We will add a figure to clarify that our proposed methods to exploit dual sparsity are not scattered, but rather developed in a progressive and iterative manner tailored to two deployment scenarios:
> - **Post-training stage** → Complete Expert Partition
> - **Inference stage** → 1T-Drop → 2T-Drop (Partial Expert Partition + Expert Reconstruction with Neuron Profiling) → Load-Aware Thresholding (for distributed inference)
>
> Additionally, S-ETP is presented entirely in the Appendix as an extended discussion.
>
> > Q2: Weak related works in neuron-level sparsity.
>
> We clarify that our work addresses three key challenges in existing neuron-level sparsity methods as discussed in Section 2.2:(1) Accuracy Sensitivity; (2) Hardware Inefficiency; (3) Activation Dependency. The reviewer-mentioned paper, CATS, in fact reflects the limitations of (1) Accuracy Sensitivity and (2) Hardware Inefficiency. Moreover, mitigating activation dependency is only a secondary benefit of our method.
>
> > Q3: All evaluations are on short-context MCQ benchmarks.
>
> Our evaluation also included GSM8K for mathematical reasoning. To further address the concern, we conduct experiments on more challenging generation and long-context benchmarks. Across these tasks, our method also achieves comparable accuracy to the baseline. Please refer to our response to [ Q2 of Reviewer etyj ] for detailed experimental description.
> |Model|Method|HumanEval|MATH500|RULER-4K|RULER-8K|RULER-16K|
> |-|-|-|-|-|-|-|
> |DeepSeek-V2-Lite-Chat|No Drop|48.2|22.4|75.7|84.4|81.6|
> |DeepSeek-V2-Lite-Chat|2T-Drop (Reconstruct)|47.7|22.1|78.8|83.6|81.0|
> |OLMoE-Instruct|No Drop|34.8|19.4|61.9|-|-|
> |OLMoE-Instruct|2T-Drop (Reconstruct)|35.4|19.8|61.7|-|-|
>
> > Q4: It's not clearly how much additional overhead the finetuning costs.
>
> We clarify that our method can be applied in two optional stages, and its overhead should be understood accordingly.
>
> 1. Post-training stage (fine-tuning):
>
> Our method seamlessly integrates into the standard post-training/instruction-tuning pipeline. Since pretrained models inherently require fine-tuning before deployment, we perform expert partitioning within this existing process **rather than adding an extra fine-tuning stage**. Thus, it introduces no additional costs, maintaining the original fine-tuning datasets and steps.
>
> 2. Inference stage (training-free, with recalibration):
>
> Independent of post-training, 2T-Drop can be applied directly at inference. It is training-free, requiring only a lightweight recalibration that collects profiling statistics without updating model parameters. The overhead is minimal, e.g. **recalibration only requires one inference pass over MMLU for profiling**, plus one verification pass on the 8 benchmarks. In our experiments, **each benchmark inference takes about 1 minute** on a single H20 GPU.
>
> Overall, our method incurs zero extra fine-tuning cost and marginal inference-time recalibration overhead.
>
> > Q5: Limited originality.
>
> As discussed in Q2 and Section 2.2, three key challenges hinder the practical adoption of existing pruning methods in modern LLMs. Rather than simply combining prior techniques, we propose a novel strategy that leverages the unique dual sparsity of MoE models to address these issues. This original insight and the resulting MoE-specific method effectively outperform prior works, demonstrating a contribution well beyond incremental improvements.
>
> > Q6: Grammar issues.
>
> We have corrected "dual sparse" to "dual sparsity."
>
> > Q7: The neuron importance profiling requires choosing both a metric and a calibration dataset, and the best metric differs across models.
>
> For the profiling metric, Section 4.3.4 reports the accuracy variance across four methods and identifies the optimal settings per model.
> The calibration dataset requires similar selection. The supplementary evaluations (see table) across various datasets and mixtures sampled from the 8 tasks show that 2T-Drop consistently improves accuracy, although the gains vary.
>
> Importantly, calibration incurs only a minute-level overhead (see Q4). Thus, tailoring the metric and dataset to a specific model introduces negligible cost rather than a substantial hidden burden. Empirically, we use MMLU as the default calibration dataset since it performs robustly across all three models.
>
> |Calibration dataset|Avg. accuracy improvement on 8 tasks over 1T-Drop|
> |-|-|
> |MMLU|+0.38|
> |Winogrande|+0.24|
> |ARC-C|+0.10|
> |Mix_sample_10K|+0.05|
> |Mix_sample_20K|+0.33|
>
> > Q8: They have not discussed the limitations in the paper.
>
> We will add a Limitations section to summarize limitations already discussed in the paper:
> 1. Section 3.1: Increasing expert partitions beyond a certain point brings marginal gains.
> 2. Section 3.3: Optimal configurations require empirical selection per model.
> 3. Section 4.3.2: Extreme speedups inevitably degrade accuracy.

---

> > ### Author Rebuttal · Reviewer_Z2HT · 2026-04-03
> >
> > Dear authors, thank you for your thoughtful comments and for providing detailed experiment results and justification. These mostly resolve my concerns, and I'll adjust the score accordingly. I especially appreciate the experimental guarding showing that this method works for more challenging generation and long-context benchmarks as well.

---

> > > ### Author Response · Authors · 2026-04-03
> > >
> > > Thank you very much for your positive feedback and for your willingness to adjust the score accordingly.
> > >
> > > In the revised manuscript, we will further improve the part of  experiments on more challenging generation and long-context benchmarks, and incorporate it in an appropriate manner to make the evidence and discussion clearer and more complete.
> > >
> > > If you have any additional questions or comments, we would be happy to discuss them further.

---

### Decision · Program_Chairs · 2026-04-30

**Decision:**

Accept (regular)

**Comment:**

This paper proposes a unified framework for exploiting tensor-level and neuron-level sparsity in Mixture-of-Experts models, introducing complete expert partition for post-training and dual-threshold token-expert dropping with neuron reconstruction for inference. The approach is validated across multiple MoE architectures with consistent accuracy retention and measurable inference speedups.

Several concerns remain after the rebuttal. The end-to-end speedups are modest (roughly 1.07--1.13x), and the manuscript does not sufficiently demonstrate that the added system complexity --- multiple threshold schemes, calibration steps, and neuron profiling --- is justified by the practical gains. The complete expert partition component lacks a clear theoretical distinction from simply training a fine-grained MoE architecture from the start; a formal analysis of why post-hoc partitioning of coarse-grained experts yields gains beyond native fine-grained designs would strengthen the contribution. Threshold selection and neuron importance profiling both require per-model empirical tuning with no automated selection mechanism, raising questions about transferability to new architectures. Significant supporting evidence --- including evaluations on generation tasks, long-context benchmarks, and newer MoE models --- appeared only in the rebuttal and has not yet been incorporated into the manuscript. The presentation would also benefit from consolidation, as the number of interleaved techniques and notation systems makes the paper harder to follow than necessary. A stronger revision should integrate the additional experiments, provide theoretical grounding for the partition strategy, and streamline the exposition.